# *Mycobacterium tuberculosis* modulates phosphorylation of host ATP6V1E1 to promote intracellular survival

Jianxia Chen[1,11], Fen Tang[2,11], Lianhua Qin[3,11], Weijun Fang[3], Liru Guan[3], Xiangyang Wu[1], Haohao Li[1], Yongjia Duan[3], Fei Wang[3], Cheng Peng[3], Zhonghua Liu[3], Jie Wang[3], Xiaochen Huang[3], Lin Wang[3,4], Hua Yang[3,4], Li Wang[5], Wei Sha[3,5], Xia Cai[6], Liang-Dong Lyu[7,8] ✉, Haipeng Liu[4,9] ✉, Feng Liu[10] ✉, Baoxue Ge[1,3,4] ✉ & Ruijuan Zheng[3,4] ✉

Intracellular pathogens such as *Mycobacterium tuberculosis* (Mtb) can promote their survival within infected cells by preventing lysosomal acidification. Here, we report that Mtb secretes a protein (Rv1184, or acyltransferase Chp2) that inhibits lysosomal acidification by targeting the host vacuolar ATPase (V-ATPase). We show that phosphorylation of the V-ATPase E1 subunit (ATP6V1E1) at Tyr56/57 suppresses lysosomal acidification through inhibition of V-ATPase assembly. Further investigation reveals that tyrosine kinase BMX promotes phosphorylation of ATP6V1E1. Strikingly, Chp2 increases BMX-dependent phosphorylation of ATP6V1E1, apparently by directly binding ATP6V1E1 and facilitating its interaction with BMX. Furthermore, inhibition of BMX impairs Mtb growth within macrophages and in mice. Thus, our work reveals a mechanism for the regulation of lysosomal acidification and suggests lysosomal acidification modulation as a potential approach for host-directed therapy against Mtb.

*Mycobacterium tuberculosis* (Mtb) has become a major killer among infectious diseases[1,2]. In 2024 global tuberculosis report, approximately 10.8 million people develop tuberculosis (TB) and 1.25 million deaths are caused by TB[3]. Mtb droplets are routinely phagocytosed by host alveolar macrophages to form phagosomes, and phagosomes undergo a series of maturation steps including fuse with lysosomes and thus develop into an increasingly acidified phagolysosomes, resulting in the effective degradation of Mtb, which is an essential process for the elimination of Mtb[4].

As an extremely successful intracellular pathogen, Mtb can survive and persist within macrophages by suppressing phagosome maturation to avoid lysosomes[5-7] hydrolases[8]. Mtb has evolved several

[1]Clinical and Translational Research Center, Shanghai Key Lab of Tuberculosis, Shanghai Pulmonary Hospital, Tongji University School of Medicine, Shanghai, China. [2]School of Medicine, Shanghai University, Shanghai, China. [3]Shanghai Key Laboratory of Tuberculosis, Shanghai Pulmonary Hospital, Tongji University School of Medicine, Shanghai, PR China. [4]Key Laboratory of Pathogen-Host Interaction, Ministry of Education, Department of Microbiology and Immunology, Tongji University School of Medicine, Shanghai, PR China. [5]Department of Tuberculosis, Shanghai Pulmonary Hospital, Tongji University School of Medicine, Shanghai, China. [6]Biosafety Level 3 Laboratory & Shanghai Medical College, Fudan University, Shanghai, PR China. [7]Key Laboratory of Medical Molecular Virology of the Ministry of Education/Ministry of Health, Department of Medical Microbiology and Parasitology, School of Basic Medical Sciences, Fudan University, Shanghai, China. [8]Shanghai Clinical Research Center for Tuberculosis, Shanghai Key Laboratory of Tuberculosis, Shanghai Pulmonary Hospital, Shanghai, China. [9]Central Laboratory, Shanghai Pulmonary Hospital, Tongji University School of Medicine, Shanghai, PR China. [10]Department of Otolaryngology-Head and Neck Surgery, Otolaryngology Institute of Shanghai JiaoTong University, Shanghai Sixth People's Hospital Affiliated to JiaoTong University Medical School, Shanghai, China. [11]These authors contributed equally: Jianxia Chen, Fen Tang, Lianhua Qin. ✉e-mail: ld.lyu@fudan.edu.cn; haipengliu2013@163.com; fenglew@sjtu.edu.cn; gebaoxue@sibs.ac.cn; zhruijuan923@163.com

strategies to suppress each stage of phagosome maturation including modulation of phagosome fusion and inhibition of phagosome acidification. Accumulating evidence indicate that mycobacterial cell wall components, such as lipoarabinomannan (LAM)[9], phosphatidylinositol mannoside (PIM)[10], and secretion proteins such as EsxG/EsxH[11], PknG[12], SapM[12] and EsxA[13], inhibit phagosome maturation through interfering with fusion processes, including phagosome-endosome fusion or phagosome-lysosome fusion. However, the mechanisms by which Mtb prevents lysosomal acidification to suppress phagosome maturation remain largely unclear.

Lysosomal acidification is fundamentally important for the degradation of macromolecules, and its dysfunction is associated with cancer, aging and neurodegenerative diseases[14–17]. Lysosomal acidification is regulated by the coordinated function of multiple ion channels and, most remarkably, vacuolar ATPase (V-ATPase), a macromolecular complex responsible for pumping protons (H + ) into lysosomes using metabolic energy in the form of ATP. The V-ATPase reduces intraluminal pH to the acidic range that is required for the activation of dozens of hydrolases with acidic pH optima in lysosomes[14,15]. Currently, only two studies report that Mtb prevents lysosomal acidification through modulating V-ATPase[18,19]. Mtb infection induces CISH expression, which triggers ubiquitination and degradation of the A subunit of V-ATPase, thereby shutting down the proton pump[18]. Additionally, the Mtb secreted protein PtpA disrupts the incorporation of V-ATPase into the phagosomal membrane, preventing phagosome acidification[19]. However, whether Mtb employs additional immune evasion mechanisms to achieve survival through V-ATPase remains unclear.

In the present study, by screening of Mtb secretory proteins that inhibit the lysosomal acidification, we observe that Mtb acyltransferase (Chp2, Rv1184)[20] inhibits lysosomal acidification through V-ATPase. Moreover, we uncover that E1 subunit of V-ATPase (ATP6V1E1) is an essential regulator for lysosomal acidification and phosphorylation of ATP6V1E1 mediated by BMX inhibits V-ATPase assembly to suppress lysosomal acidification, regulating the anti-mycobacterial immune response. However, Mtb Chp2 enhances the binding of ATP6V1E1 to BMX, increasing the BMX-mediated phosphorylation of ATP6V1E1 to promote Mtb survival.

## Results

### Chp2 inhibits lysosomal acidification through V-ATPase

Lysosomal acidification is essential for the clearance of Mtb[21]. To explore which *Mtb* protein inhibits host lysosomal acidification, 201 plasmids encoding Mtb-secreted proteins or lipoproteins were transfected into HEK293T cells, and their effects on lysosomal acidification were measured using LysoTracker staining, an acidotrophic fluorescent dye that accumulated in acidic organelles[22]. Chp2, an acyltransferase of Mtb, was the leading first protein of inhibiting lysosomal acidification (Fig. 1a-d). Chp2 was detected in both lysates and supernatants from cultures of H37Rv by using anti-Chp2 antibody, but not in the cultures of the H37RvΔChp2 (Supplementary Fig. 1a, b), conforming that Chp2 is a secreted protein. In Mtb-infected macrophages, H37RvΔChp2 markedly promoted the lysosomal acidification, and complementation of H37RvΔChp2 with Chp2 restored the its inhibition effect on lysosomal acidification (Fig. 1e, f, Supplementary Fig. 1c, d), validating that Chp2 inhibited lysosomal acidification. Furthermore, we assessed the co-localization of H37Rv, H37RvΔChp2, and the complemented strain with lysosomes using LAMP1 and evaluated lysosomal acidification using LysoTracker. All strains showed lysosomal co-localization without significant differences (Supplementary Fig. 1e, f). In parallel, H37RvΔChp2 notably enhanced lysosomal acidification compared to H37Rv, whereas complementation restored the acidification blockade (Supplementary Fig. 1g, h), demonstrating that Mtb inhibits lysosomal acidification via Chp2. V-ATPase is essential for lysosomal acidification, as it is the sole transporter that pumps protons

into the organelle[23]. The use of bafilomycin A1 (Baf-A1)[20], a specific inhibitor of V-ATPase, to inhibit lysosomal acidification and perform functional rescue experiments is a well-established and widely adopted strategy to validate whether a molecular mechanism acts through V-ATPase-mediated acidification. Given this, macrophages were pretreated with Baf-A1 prior to infection with H37Rv and H37RvΔChp2 and then stained with LysoTracker to assess lysosomal acidification. As expected, Baf-A1 treatment significantly reduced lysosomal acidification in H37Rv-infected macrophages, confirming that the Mtb-induced acidification is mediated by V-ATPase. Furthermore, V-ATPase inhibition abolished the difference in acidification levels observed between H37Rv and H37RvΔChp2 (Fig. 1g, h, Supplementary Fig. 1i), demonstrating that Chp2 inhibited lysosomal acidification via V-ATPase.

### Chp2 promotes Mtb survival through V-ATPase

Initially, we found no significant difference between the in vitro growth rate of wild-type H37Rv and H37RvΔChp2 (Supplementary Fig. 2a). We then investigated the functional role of Chp2 in the clearance of Mtb. Using a colony-forming unit (CFU) assay, we found similar intracellular bacterial loads for H37Rv and H37RvΔChp2 at 2 hours post-infection in peritoneal macrophages, suggesting Chp2 has no effects on the phagocytosis of Mtb by macrophages. However, at 24 to 96 hours post-infection, deletion of Chp2 dramatically reduced the intracellular bacteria CFU count compared to H37Rv controls (Fig. 2a, b). In addition, deletion of Chp2 also reduced Mtb survival in BMDMs (Supplementary Fig. 2b). As a weak base precursor, $NH_4Cl$ inhibits lysosomal acidification by diffusing into acidic compartments and neutralizing them[24]. Therefore, it can serve as a positive control for inducing a lysosomal impairment phenotype. The treatment of macrophages with Baf-A1 (Fig. 2c, d) or $NH_4Cl$ (Fig. 2e, f), eliminated the difference in intracellular survival between H37Rv and H37RvΔChp2. In vivo, the mice were infected with H37Rv, H37RvΔChp2 and H37Rv(ΔChp2+Chp2) for 4 weeks (Supplementary Fig. 2c). Chp2 deletion dramatically reduced histopathological changes in lung tissues (Fig. 2g, h). The multiplex immunohistochemistry demonstrated a significant reduction in both innate and adaptive immune cells, including monocytes/macrophages, neutrophils, T cells and B cells (Supplementary Fig. 2d, e). Consistently, Chp2 deletion comparably had a much lower bacterial loads in the lungs of mice (Fig. 2i, j). Complementation of H37RvΔChp2 with Chp2 restored the lung tissue pathological damage and bacterial burden (Fig. 2g-j), suggesting Chp2 is a virulence factor and exacerbated Mtb infection. The reduction in overall immune cell infiltration is a in vivo consequence of the enhanced bacterial clearance mediated by the Chp2 deletion. This superior early control leads to a lower overall bacterial burden and a subsequently attenuated inflammatory response in the lungs. Furthermore, mice were intraperitoneally injected with Baf-A1 prior to infection with H37Rv or H37RvΔChp2 for 3 days, followed by the weekly Baf-A1 injections for 4 weeks (Supplementary Fig. 2f). In H37Rv-infected mice, Baf-A1 treatment significantly increased the bacterial load in the lungs compared to the untreated group, demonstrating that the clearance of Mtb in vivo is dependent on V-ATPase. Furthermore, V-ATPase inhibition by Baf-A1 abolished the difference in lung bacterial burden between H37Rv and H37RvΔChp2 infection (Fig. 2k), indicating that Chp2 enhances bacterial survival in mice by targeting the V-ATPase. Together, Chp2 promoted the Mtb survival both within macrophages and in vivo through V-ATPase mediated lysosomal acidification.

### Chp2 interacts with subunit E1 of human V-ATPase

The V-ATPases are large, multi-subunit protein complexes essential for cellular acidifications[25]. To explore how Chp2 regulates lysosomal acidification, we investigated its interaction with V-ATPase subunits by immunoprecipitation. We found E1 subunit of V-ATPase (ATP6V1E1) in the Chp2 precipitates (Fig. 3a–c, Supplementary Fig. 3a).

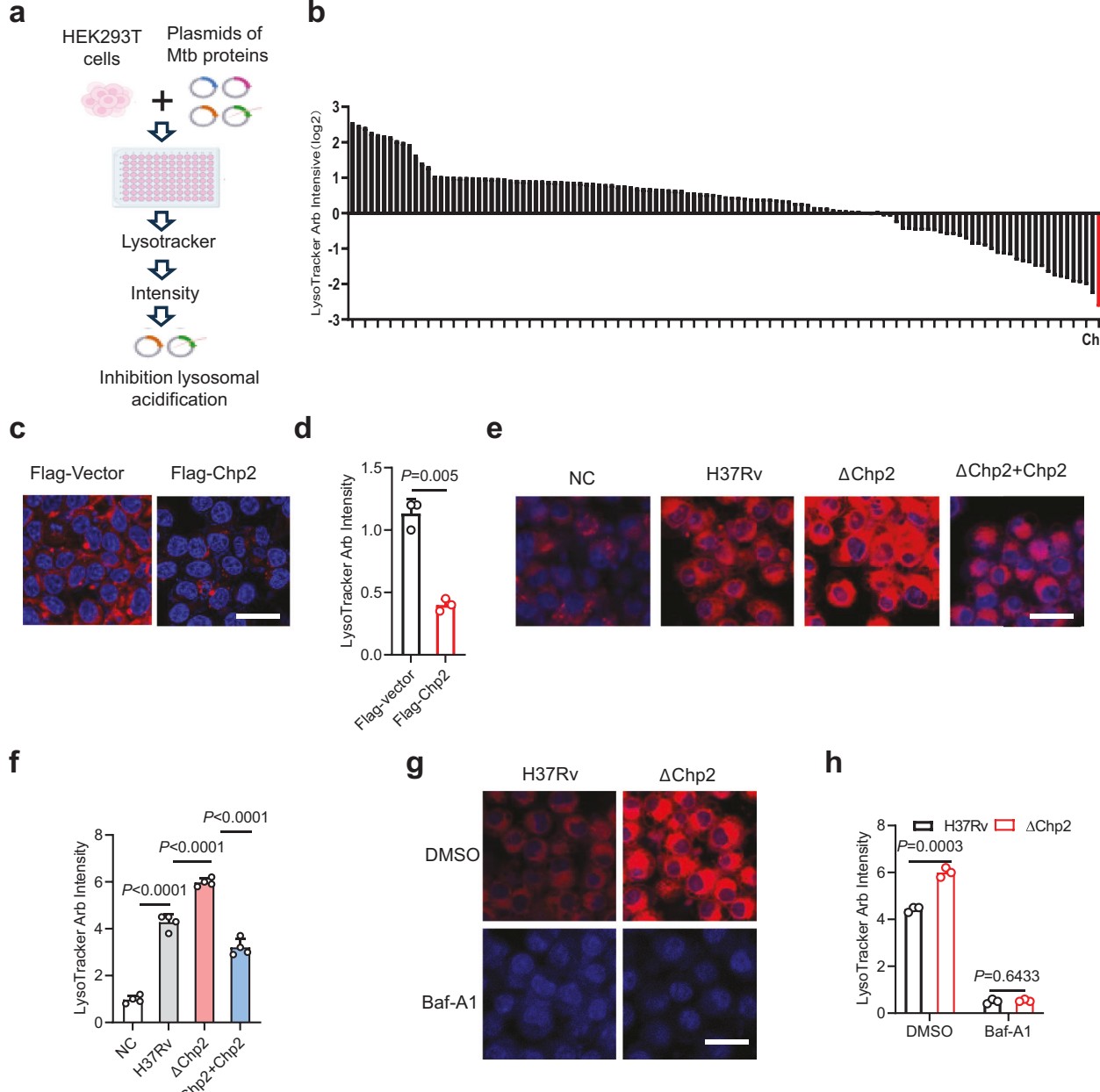

**Fig. 1 | Chp2 inhibits lysosomal acidification. a** Schematic diagram of the experimental procedure used to identify the Mtb proteins that inhibited lysosomal acidification in HEK293T cells. **b** Screening Mtb proteins for regulation of lysosomal acidification. HEK293T cells were transfected with plasmids overexpressing Mtb proteins for 48 hours, followed by LysoTracker staining. LysoTracker quantifications data were generated using high-content analysis (Cellomics). **c** Representative images of Lysotracker staining in HEK293T cells overexpressing either an empty vector or Flag-tagged Chp2. Scale bar, 20 μm. **d** Quantification of LysoTracker fluorescence intensity. (mean ± SEM). **e** Representative images of LysoTracker staining in primary peritoneal macrophages infected with the following strains:

H37Rv, H37RvΔChp2, H37Rv(ΔChp2+Chp2) (MOI = 5) for 4 h. Scale bar, 20 μm. **f** Quantification of LysoTracker fluorescence intensity (mean ± SEM), P = 2.05401E-06 (H37Rv), P = 9.64311E-05 (ΔChp2), P = 8.07779E-06 (ΔChp2+Chp2). **g** Representative images of LysoTracker staining in primary peritoneal macrophages pretreated with or without Baf-A1(1 μM) for 1 h, followed by infection with H37Rv or H37RvΔChp2 (MOI = 5) for 4 h. Scale bar, 20 μm. **h** Quantification of LysoTracker fluorescence intensity (mean ± SEM). Data in (**a**–**h**) are representative of one experiment with at least three independent biological replicates (**d,h**, n = 3; f, n = 4). Two-tailed unpaired Student's t test (d, f and h) was used for statistical analysis. Elements in Fig. 1a was created in BioRender[50].

Immunofluorescence assay confirmed the colocalization of Chp2 with ATP6V1E1 (Supplementary Fig. 3b), and the in vitro precipitation assay further demonstrated that recombinant Chp2 directly binds to ATP6-V1E1(Fig. 3d, e). Moreover, the surface plasmon resonance (SPR) assay showed that Chp2 had a strong affinity for ATP6V1E1 with an affinity constant of 21.3 nM (Fig.3f). ATP6V1E1 is a subunit that forms a peripheral stalk with a stator function for stabilizing the V1 domains of V-ATPase complexes. Among 11 subunit isoforms of V1 domain,

ATP6V1E1 had the highest intensity of lysotracker staining, suggesting its potential role in regulating acidification (Supplementary Fig. 3c, Fig. 3g, h). To better characterize the function role of ATP6V1E1 in Mtb infection in vivo, we therefore generated *Atp6v1e1*-deficient mice by CRISPR/Cas9-mediated genome editing (Supplementary Fig. 3d-f). The homozygous deletion of *Atp6v1e1* results in embryonic lethality, which is consistent with the Mouse Genome Informatics (MGI) database. The *Atp6v1e1* heterozygous mice appear normal and do not exhibit

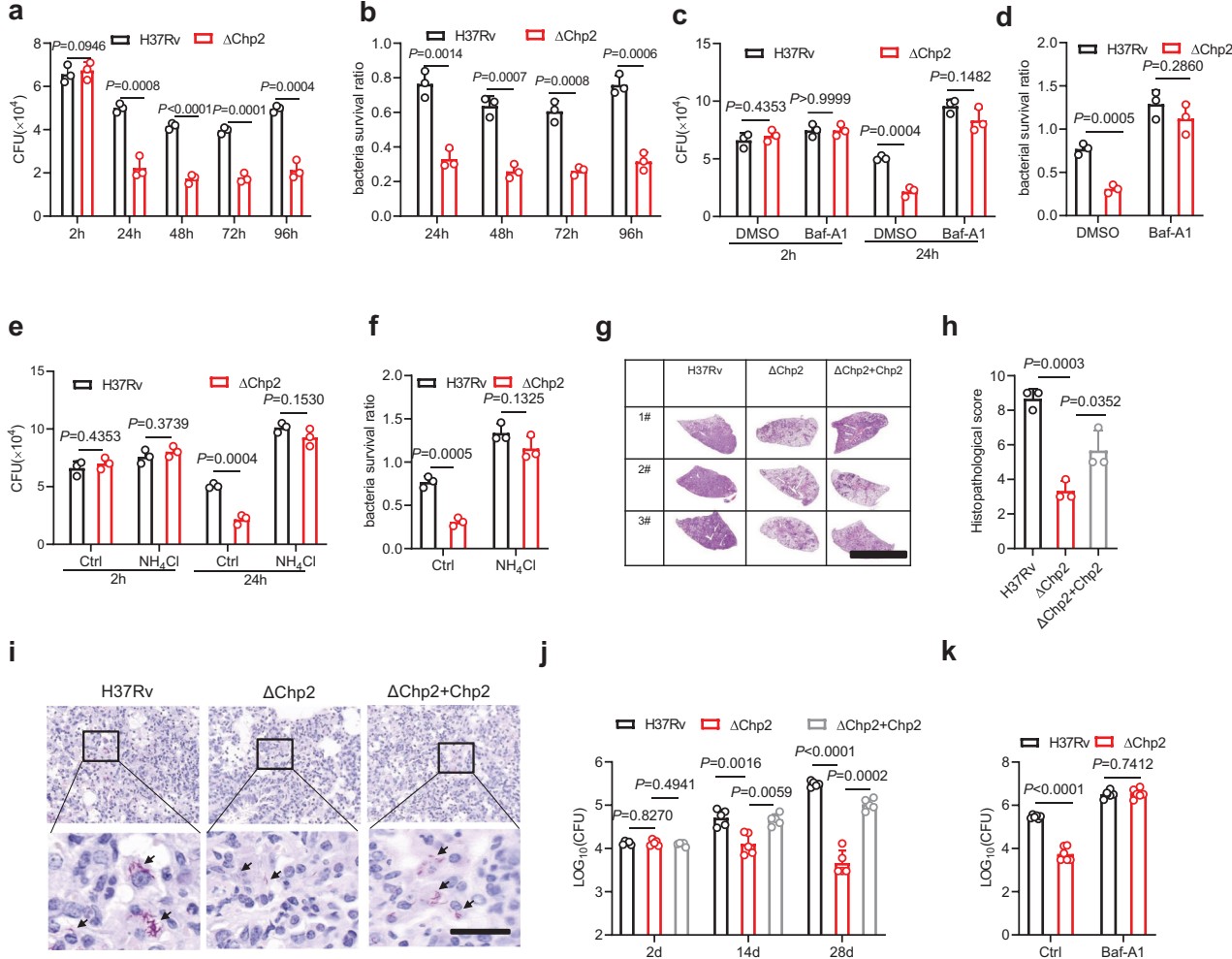

**Fig. 2 | Chp2 promote intracellular survival of Mtb. a–f** Intracellular CFU and bacterial survival in mice peritoneal macrophages infected with H37Rv or H37RΔChp2 for the indicated times (a and b) (MOI = 5) or then treated with Baf-A1(1 μM) (c and d) or NH₄Cl (20 μM) (e and f) for 1 h prior to infection (mean ± SEM), a, *P* = 8.25045E-05(48 h). **g** H&E staining of lung tissues from C57BL/6 mice infected with H37Rv, H37RvΔChp2 or H37Rv(ΔChp2+Chp2) by intranasal infection ( ~ 200 CFU) for 4 weeks. #1-3 indicated three representative lung tissues. Scale bar, 1 mm. **h** Quantification of the histopathology score in the lungs of mice infected with Mtb as in (**g**) (mean ± SEM). **i** Acid-fast staining of lung tissues of mice infected with Mtb as in (**g**). **j** Bacterial CFUs in the lungs of mice infected with Mtb as in (**g**) (j, mean ± SEM, *N* = 5 mice). *P* = 6.35756E-07 (28 d). Scale bar, 100 μm. **k** Bacterial CFUs in the lung tissues of mice injected intraperitoneally Baf-A1(10 mg/kg) before undergoing H37Rv or H37RvΔChp2 infection for 3 days and then were treated once weekly for 4 weeks (mean ± SEM, *N* = 6 mice). *P* = 1.57695E-07 (Ctrl). Data in (**a–k**) are representative of one experiment with at least three independent biological replicates (**a–f,h**, *n* = 3). Two-tailed unpaired Student's *t* test (**a–f,h,k**) was used for statistical analysis. Two-sided Mann-Whitney U test (**j**) was used for statistical analysis.

lysosomal storage disorder phenotypes in the lung (Supplementary Fig. 3g). Therefore, we used *Atp6v1e1* heterozygous mice in our study. Upon Mtb infection, silencing of *Atp6v1e1* with small interfering RNA (siRNA) decreased Mtb induced lysosomal acidification (Supplementary Fig. 3h–j). Deletion of *Atp6v1e1* resulted in a decreased Lyso-Tracker staining (Fig. 3i, j). In addition, using pHrodo™-dextran, a pH-sensitive probe that exhibits a strong increase in fluorescence specifically upon acidification, also shown an attenuated oregon-green dextran (OGD) staining (Supplementary Fig. 3k), and an increased lysosomal pH in macrophages infected with Mtb (Supplementary Fig. 3l). Intriguingly, the deletion of *Atp6v1e1* markedly increased the intracellular survival of Mtb (Fig. 3k, l). Treatment with NH₄Cl attenuated *Atp6v1e1* deletion mediated enhancement of Mtb survival in macrophages (Fig. 3m). In vivo, the *Atp6v1e1⁺ᐟ⁻* mice infected with Mtb showed an increased bacterial burden in their lungs compared to the wild type mice (Fig. 3n, Supplementary Fig. 3m, n) and much severer pathological impairment (Fig. 3o, p). These results suggested

ATP6V1E1 was an essential regulator for lysosomal acidification to defense against Mtb infection.

Given that Chp2 regulates lysosomal acidification via the V-ATPase, we hypothesized that it might specifically target the E1 subunit of V-ATPase. To test this, macrophages from WT and *Atp6v1e1⁺ᐟ⁻* mice were infected with H37Rv or H37RvΔChp2 and then stained with LysoTracker. The results were revealed that deletion of *Atp6v1e1* attenuated H37RvΔChp2-induced enhancement of lysosomal acidification in the infected macrophages (Fig. 3q, r, Supplementary Fig. 3o). Furthermore, deletion of *Atp6v1e1* eliminated H37RvΔChp2-mediated inhibition of Mtb intracellular survival in macrophages (Fig. 3s). After treating cells with NH₄Cl, no difference in Mtb intracellular survival was observed in WT or *Atp6v1e1⁺ᐟ⁻* macrophages infected with H37Rv or H37RvΔChp2 (Fig. 3t), suggesting Chp2 inhibits lysosomal acidification to promote intracellular survival of Mtb through ATP6V1E1. Moreover, we infected *Atp6v1e1⁺ᐟ⁻* mice with H37Rv or H37RvΔChp2 to further validate in vivo relevance of Chp2 and

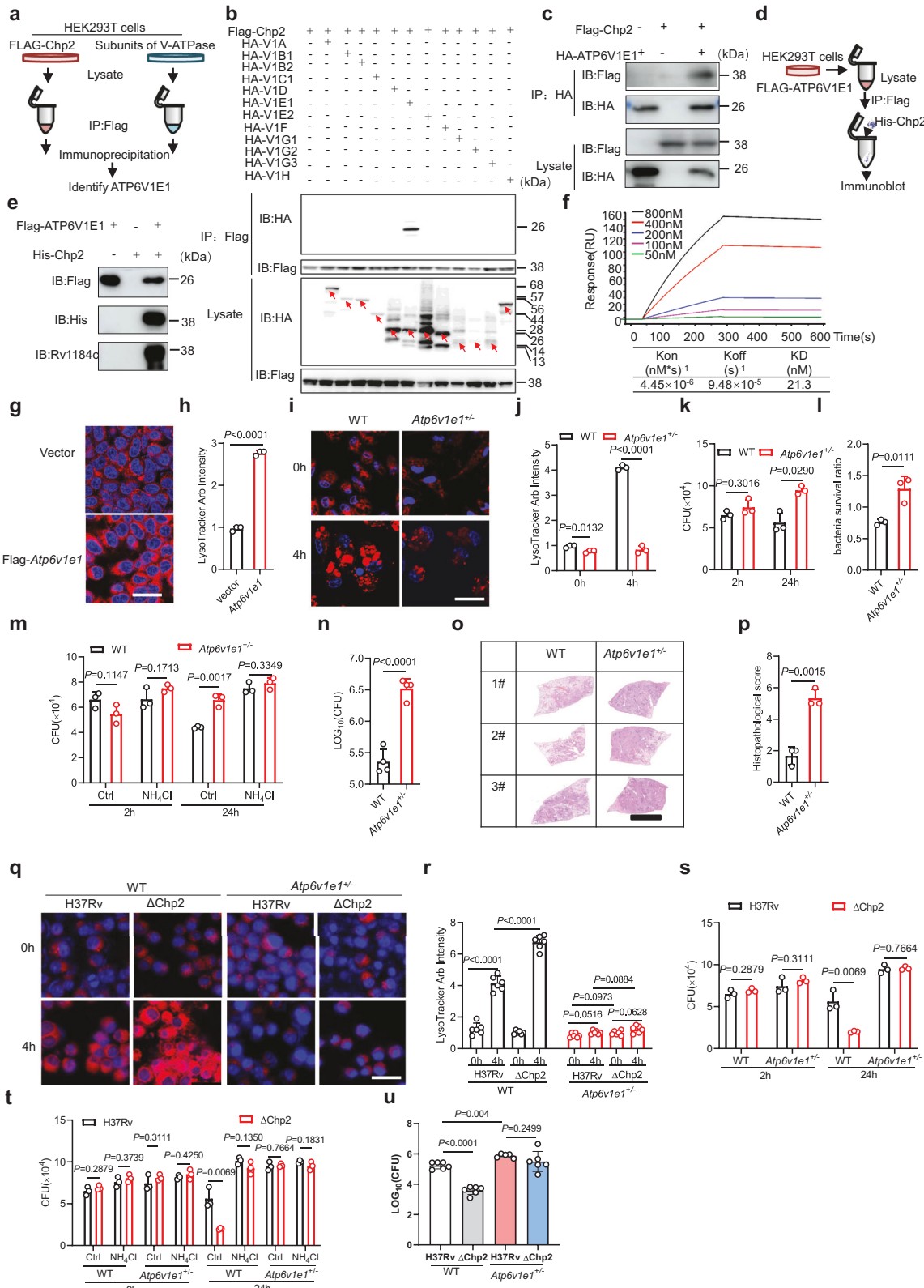

ATP6V1E1. Knockout of *Atp6v1e1* markedly increased bacterial burden of the Mtb H37Rv-infected mice, and abolished the increased bacterial burden by Chp2 in lung tissues of the Mtb-infected mice (Fig. 3u, Supplementary Fig. 3p). To assess the direct pathogenic potential of the Chp2 protein, we administered purified recombinant Chp2 intravenously or intranasally to naive C57BL/6 mice over a three-week period. Histopathological analysis of H&E-stained lung sections revealed that, compared to control groups treated with PBS, Chp2 alone did not induce significant pulmonary damage (Supplementary Fig. 3q, r). Given the lack of direct pathology, the significant impact of Chp2 on bacterial survival and pathogenesis likely depends on ATP6V1E1. Together, our results demonstrate that Chp2 promotes intracellular survival of Mtb in macrophages and in vivo by targeting ATP6V1E1 to inhibit lysosomal acidification.

**Fig. 3 | Chp2 interacts with ATP6V1E1. a** Schematic of the experimental workflow for identifying Chp2-interaction proteins in HEK293T cells. **b,c** Co-immunoprecipitation (Co-IP) assay in HEK293T cells. Cells were transfected for 48 h with plasmids encoding Flag-Chp2 and individual HA-tagged subunits of the V-ATPase. Cell lysates were immunoprecipitated with an anti-Flag antibody and immunoblotted with the indicated antibodies. Chp2. **d** Schematic diagram of the in vitro pull-down assay. **e** In vitro pull-down assay. FLAG beads bound to Flag-ATP6V1E1 from transfected HEK293T cells were incubated with recombinant Chp2 protein. After washing, the beads were immunoblotted to detect interaction. **f** Surface plasmon resonance (SPR) assay of the direct interaction of ATP6V1E1 with Chp2. The calculated equilibrium dissociation constant (KD) was 21.3 nM. **g** Representative Lysotracker staining of HEK293T cells overexpressing Flag-ATP6V1E1 for 48 h. Scale bar, 20 μm. **h** Quantification of LysoTracker fluorescence intensity. (mean ± SEM), $P = 2.81076E{-}06$. **i** Representative LysoTracker staining of macrophages isolated from WT and $Atp6v1e1^{+/-}$ mice infected with H37Rv for 4 h (MOI = 5). Scale bar, 20 μm. **j** Quantification of LysoTracker fluorescence intensity. (mean ± SEM). $P = 6.46035E{-}06$ (4 h). **k–m** Intracellular CFU and bacterial survival in macrophages from WT and $Atp6v1e1^{+/-}$ mice infected with H37Rv (MOI = 5) for the

indicated times (**k, l**) or treat with or without $NH_4Cl$ (20 μM) for 1 h prior to infection (**m**) (mean ± SEM). **n** Bacterial CFUs in the lungs of WT and $Atp6v1e1^{+/-}$ mice infected with H37Rv (~200 CFU) by intranasal infection for 4 weeks (mean ± SEM, N = 4 mice), $P = 9.37349E{-}05$. **o** H&E staining of lung tissues as in (**n**). Images #1-3 show three individual mice. Scale bar, 5000 μm. **p** Histopathology score of lung sections from (**o**) (mean ± SEM). **q** Representative LysoTracker staining of macrophages from WT and $Atp6v1e1^{+/-}$ mice infected with H37Rv and H37RvΔChp2 (MOI = 5) for the indicated times. Scale bar, 20 μm. (**r**) Quantification of Lyso-Tracker fluorescence intensity (mean ± SEM). $P = 3.56017E{-}07$ (H37Rv, WT), $P = 3.11248E{-}11$ (ΔChp2, WT). (**s, t**) Intracellular CFU of WT and $Atp6v1e1^{+/-}$ macrophages infected with H37Rv or H37RvΔChp2 (MOI = 5) for the indicated times (**s**) or in the absence or presence of $NH_4Cl$ (20 μM) (**t**) (mean ± SEM). (**u**) Bacterial CFUs in the lungs of WT and $Atp6v1e1^{+/-}$ mice infected with H37Rv or H37RvΔChp2 (~200 CFU) by intranasal infection for 4 weeks. (mean ± SEM, N = 6 mice). $P = 3.80346E{-}07$ (WT). Data in (**a–u**) are representative of one experiment with at least three independent biological replicates (**h,j–m,p,s,t**, n = 3; r, n = 6). Two-tailed unpaired Student's *t* test (h, j-m and r-t) was used for statistical analysis. Two-sided Mann-Whitney U test (n, u) was used for statistical analysis.

## Phosphorylation of ATP6V1E1 inhibits lysosomal acidification by suppressing V-ATPase assembly

It has been reported that the phosphorylation of V-ATPase subunits affects lysosomal acidification. To examine whether post-translational modifications of ATP6V1E1 regulate lysosomal acidification, we analyzed the phosphorylation modification of ATP6V1E1 by phosphorylation site prediction and found four potential phosphorylation sites (Fig. 4a). We mutated these sites to alanine and overexpressed them in HEK293T cells. After Lysotracker staining, we found that a dual-site mutation at residues 56 and 57 to alanine promoted lysosomal acidification, while a dual-site mutation to glutamic acid, mimicking the phosphorylated state, inhibited lysosomal acidification. Mutations at other individual sites had no effect on lysosomal acidification (Fig. 4b, c). Sequence alignment analysis revealed that tyrosine 56 and tyrosine 57 of ATP6V1E1 were highly conserved cross species (Fig. 4d). Using phosphor-tyrosine antibody 4G10, we confirmed dual-site mutation at tyrosine 56 and 57 to alanine markedly decreased phosphorylation of ATP6V1E1 in the recombinant proteins purified both from TKB1 (Fig. 4e) or HEK293T cells (Fig. 4f). We then generated the specific antibody against the tyrosine 56 and 57 and further confirmed ATP6V1E1 was phosphorylated at $Tyr^{56/57}$ in HEK293T cells that over-expressed ATP6V1E1 and mutants (Fig. 4g). These results suggested that $Tyr^{56/57}$-dependent phosphorylation of ATP6V1E1 suppressed lysosomal acidification.

V-ATPase assembly through reversible dissociation of its V1 and V0 domains was the most important forms of V-ATPase regulation[26,27]. To test whether phosphorylation of ATP6V1E1 regulates V-ATPase assembly, western blotting was performed on membrane and cytosolic fractions isolated from HEK293T cells that overexpressed ATP6V1E1 and mutants. Since the peripheral V1 domain was only present in membranes in fully assembled V-ATPase complexes, the abundance of V1 subunits in the membrane fraction indicated the level of V-ATPase assembly[14,28]. The results were shown $Tyr^{56/57A}$ mutant of ATP6V1E1 led to a significant increase in V-ATPase assembly, while $Tyr^{56/57E}$ mutant attenuated V-ATPase assembly (Fig. 4h, i), indicating the phosphorylation of ATP6V1E1 suppresses lysosomal acidification might through inhibiting V-ATPase assembly.

Through structural analysis, we further examined the influence of phosphorylation at Tyr56/57 on the structure and functionality of ATP6V1E1 and V-ATPase. The findings revealed that ATP6V1E1 and ATP6V1G1 form two straight, parallel α-helical structures, which subsequently organized into a robust, zipper-like arrangement. Notably, Tyr56/57 resided in the central region of the helices (Fig. 4j). Alpha-Fold3 simulations indicated that phosphorylation of Tyr57 facilitated an electrostatic interaction with Arg48 in ATP6V1G1, contributing to enhanced structural stability and reinforcing the overall zipper-like

conformation (Fig. 4k, l). Together, the phosphorylation of the E1 subunit may induce a conformational change in the V-ATPase peripheral stalk, regulating its reversible assembly.

## BMX regulates ATP6V1E1 phosphorylation to inhibit lysosomal acidification and promote Mtb survival

To investigate the regulatory mechanism of ATP6V1E1 phosphorylation, we screened a short hairpin RNA (shRNA) library containing 38 non-receptor tyrosine kinases to identify the kinase responsible for phosphorylating ATP6V1E1.Transfection of HEK293T cells with shRNA targeting bone marrow tyrosine kinase on chromosome X (BMX), a TEC family kinase associated with numerous pathological pathways in cancer cells[29,30], substantially reduced the phosphorylation of ATP6V1E1 at Tyr56/57 (Fig. 5a). To confirm BMX as a kinase of ATP6V1E1, we performed an in vitro phosphorylation assay. The recombinant protein of ATP6V1E1 and $ATP6V1E1^{Y56/57A}$ purified from HEK293T cells, were incubated with purified GFP-BMX protein in optimized kinase buffer. The results demonstrated that ATP6V1E1 was phosphorylated by BMX at $Tyr^{56/57}$ site (Fig. 5b). Upon Mtb infection, silencing of *Bmx* (Supplementary Fig. 4a, Fig. 5c) or inhibition of BMX using the specific kinase inhibitor BMX-IN-1(Fig. 5d), markedly reduced the $Tyr^{56/57}$ phosphorylation of ATP6V1E1 in macrophages. Consistently, deletion of *Bmx* resulted in decreased phosphorylation of ATP6V1E1 at $Tyr^{56/57}$ in macrophages infected with Mtb (Fig. 5e), suggesting that Mtb regulates phosphorylation of ATP6V1E1 via BMX. To determine if Bmx-mediated phosphorylation of ATP6V1E1 impacts the assembly of the V-ATPase complex at the lysosomal membrane, we analyzed the lysosomal fractions isolated from BMX-IN-1 treated vs. untreated macrophages, as well as from $Bmx^{+/-}$ vs. wild-type macrophages. The presence of key V1subunit ATP6V1E1 and V0 subunit ATP6V0d1 in the lysosomal membrane was assessed by immunoblotting. The results showed that Bmx inhibition either by pharmacologically or genetically alter the abundance of V-ATPase subunits at the lysosomal membrane (Fig. 5f). This suggests that Bmx inhibits V-ATPase assembly likely via changing the phosphorylation status of ATP6V1E1 subunit. Silencing of *Bmx* markedly increased the lysosome acidification in macrophages infected with Mtb as revealed by Lyso-Tracker staining (Fig. 5g, h, Supplementary Fig. 4b) and dramatically reduced the intracellular survival of Mtb as determined by the CFU assay (Fig. 5i). Moreover, in $Atp6v1e1^{+/-}$ macrophages, treatment with BMX inhibitor abrogated the BMX-mediated increase in intracellular bacteria load (Fig. 5j). We also detect Mtb intracellular survival in WT and $Bmx^{+/-}$ macrophages infected with H37Rv and H37RvΔChp2. Deletion of either *Bmx* robustly reduced Mtb intracellular survival only in H37Rv-infected murine peritoneal macrophages and eliminated the difference in intracellular survival between H37Rv and H37RvΔChp2

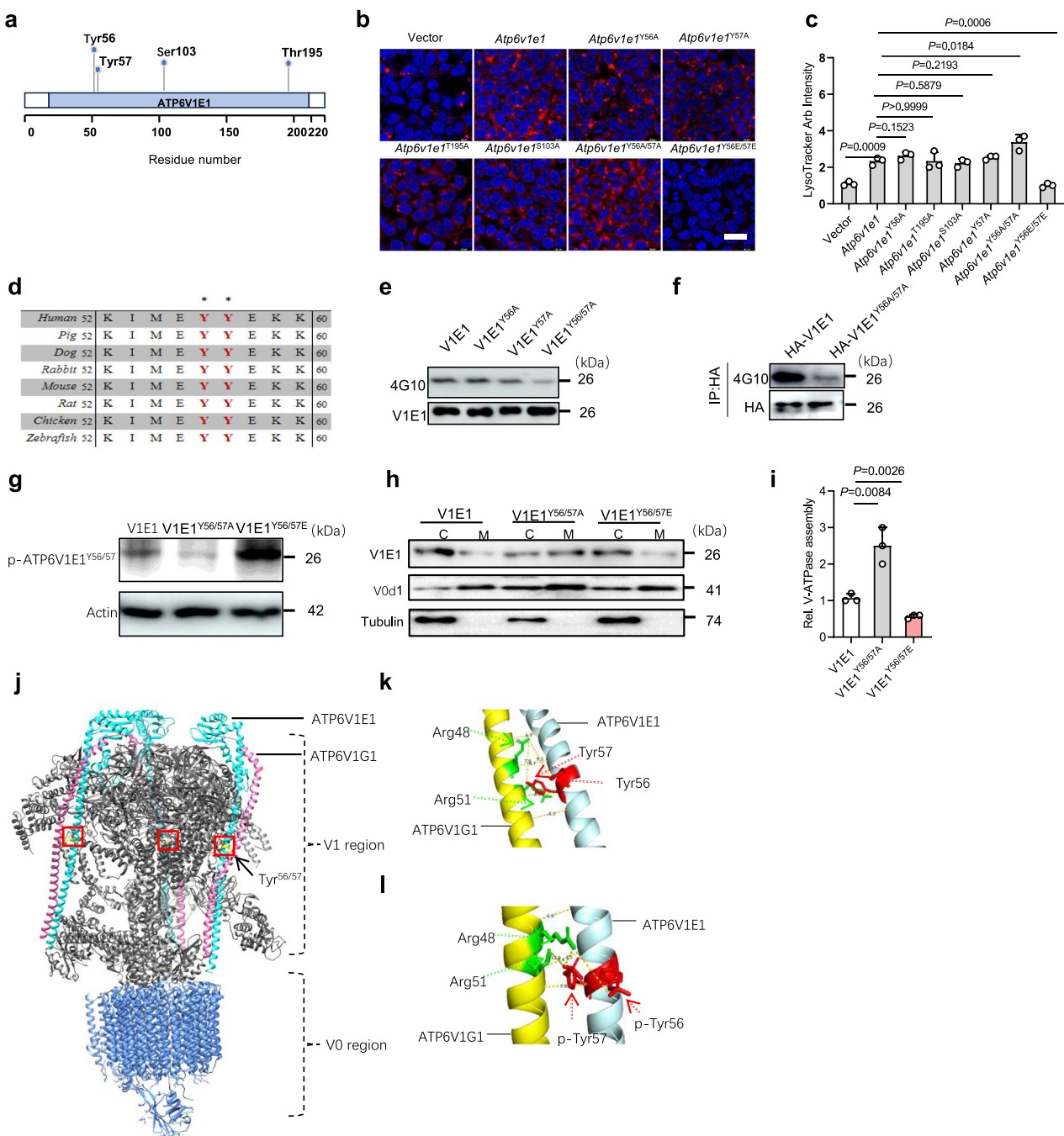

**Fig. 4 | Phosphorylation of ATP6V1E1 on Tyr56 and Tyr57 inhibits lysosomal acidification. a** Prediction of potential phosphorylation sites in the ATP6V1E1 by a web-based phosphorylation site prediction databases PhosphoSitePlus® PTM. **b** Representative LysoTracker staining of HEK293T cells overexpressing HA-tagged ATP6V1E1 and mutants. **c** Quantification of LysoTracker fluorescence intensity. (mean ± SEM). **d** Sequence alignment of the region surrounding tyrosine residues 56 and 57 (highlighted in red) of ATP6V1E1 across different species. **e,f** Immunoblotting of recombinant protein for ATP6V1E1 and mutants purified from TKB1 cells (**e**) and HEK293T cells transfected with plasmids for 48 h (f) by 4G10 antibody. **g** Immunoblotting of lysates of HEK293T cells transfected with plasmids encoding HA-ATP6V1E1 and mutants using a custom phospho antibody p-ATP6V1E1$^{Y56/Y57}$. **h** HEK293T cells were transfected with plasmids encoding HA-ATP6V1E1 and mutants for 48 h, followed by subcellular fractionation. Western blotting was performed on the cytosolic (C) and membrane (M) fractions using the indicated antibodies. E1 is a $V_1$ subunit and indicates the abundance of the $V_1$ domain in each fraction. d1 is a $V_O$ subunit and was used as a membrane loading control, while tubulin was used as a cytosolic loading control. The relative abundance of $V_1$ in the membrane fraction versus the cytosolic fraction is a measure of V-ATPase assembly. (**i**) Quantification of Western blots described in panel (**h**). The intensity of E1 in the membrane fraction was normalized to d1. The intensity of E1 in the cytosolic fraction was normalized to tubulin. The ratio of the normalized intensities gives the relative assembly (mean ± SEM). (**j**) The overall structure of V-ATPase (derived from PDB: 7U4T), viewed from the membrane side, with the ATP6V1E1 and ATP6V1G1 highlighted in cyan and pink, respectively. Tyr56/57 of the E1 subunit of V-ATPase is marked with a red box. **k,l** Detailed views of the interaction between ATP6V1G1 and ATP6V1E1, showing Tyr56/57 in the unphosphorylated state (**k**) and the phosphorylated state (**l**). Data in (**a**–**i**) are representative of one experiment with at least three independent biological replicates (**c,i**, $n = 3$). Two-tailed unpaired Student's $t$ test (**c, i**) was used for statistical analysis.

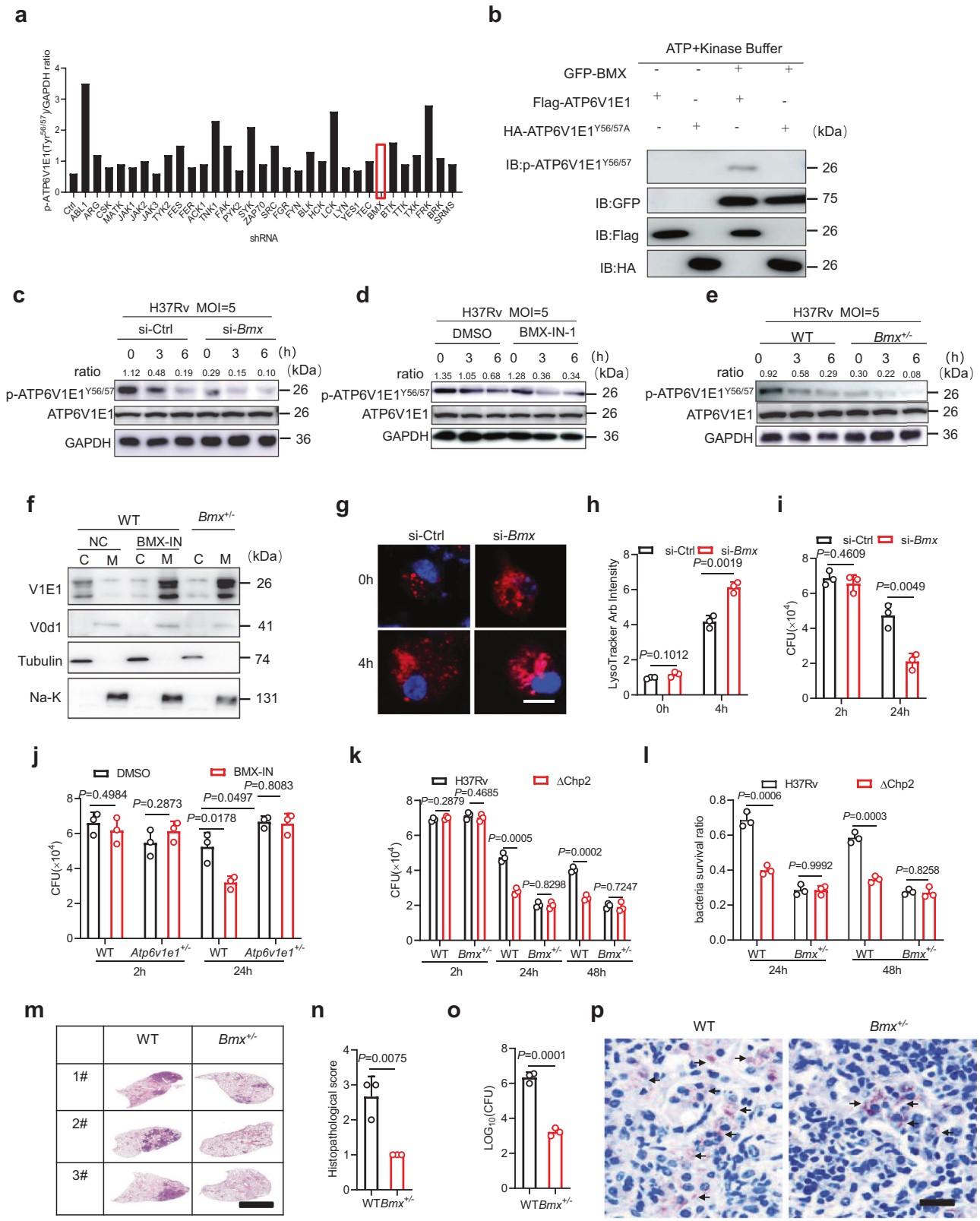

(Fig. 5k, l). Additionally, after 4 weeks Mtb infection, *Bmx*[+/-] mice exhibited a dramatically reduction in histopathological changes (Fig. 5m, n) as well as the bacterial burden in their lungs compared to the WT animals (Fig. 5o, p). Taking together, these results suggested that BMX might promote phosphorylation of ATP6V1E1 to inhibit lysosome acidification, thus promoting intracellular survival of Mtb.

## Chp2 promotes phosphorylation of ATP6V1E1 by enhancing its binding with BMX

To explore whether and how Chp2 regulates the phosphorylation of ATP6V1E1, we assessed the phosphorylation level of ATP6V1E1 in macrophages infected with H37Rv and H37RvΔChp2. Deletion of Chp2 led to a decrease in ATP6V1E1 phosphorylation (Fig. 6a). The inhibition

**Fig. 5 | BMX controls phosphorylation of ATP6V1E1. a** Quantification of western blots to screen Kinase library regulated ATP6V1E1 phosphorylation. **b** In vitro phosphorylation assay of ATP6V1E1. The proteins of GFP-BMX, Flag-ATP6V1E1 and HA-ATP6V1E1[Y56/Y57A] were overexpressed and purified in HEK293 cells. **c-e** Detection of ATP6V1E1 phosphorylation in macrophages transfected with *Bmx* siRNA (**c**), treated with BMX inhibitor (10 nM) (**d**) or in *Bmx*[+/-] macrophages (**e**), then infected with H37Rv (MOI = 5) for indicated times. ratio: p-ATP6V1E1[Y56/57]/ATP6V1E1. **f** Subcellular fractionation of macrophages from WT or *Bmx*[+/-] mice or WT macrophages under the treatment with BMX-IN (10 nM) for 1 h. Cytosolic (C) and membrane (M) fractions were immunoblotted with antibodies against the V1 domain subunit ATP6V1E1 and the V0 domain subunit ATP6V0d1. α-Tubulin and V0d1 serve as loading controls for cytosolic and membrane fractions, respectively. **g** Representative LysoTracker staining of macrophages transfected with scrambled or *Bmx*-specific siRNA and then infected with H37Rv (MOI = 5) for the indicated times. Scale bar, 10 μm. **h** Quantification of LysoTracker fluorescence intensity (mean ± SEM). **i** Intracellular CFU in macrophages transfected scrambled or *Bmx*-specific siRNA and then infected with H37Rv (MOI = 5) for the indicated times (mean ± SEM). **j** Intracellular CFU in macrophages from WT and *Atp6v1e1*[+/-] mice treated with BMX inhibitor (10 nM) and infected with H37Rv (MOI = 5) for the indicated times (mean ± SEM). **k-l** Intracellular CFU and bacterial survival in macrophages from WT or *Bmx*[+/-] mice infected with H37Rv or H37RvΔChp2 for the indicated times (MOI = 5) (mean ± SEM). (**m**) H&E staining of lung tissues from WT and *Bmx*[+/-] mice infected with H37Rv by intranasal infection ( ~ 200 CFU) for 4 weeks. (mean ± SEM, N = 3 mice) #1-3 indicate three representative lung tissues. Scale bar, 5000 μm. **n** Quantification of the histopathology score in (**m**) (mean ± SEM). **o** Bacterial CFUs in the lungs of WT and *Bmx*[+/-] mice in (**m**) (mean ± SEM). **p** Acid-fast staining of the lung tissues of WT and *Bmx*[+/-] mice in (**m**). Scale bar, 100 μm. Data in (**a-m**) are representative of one experiment with at least three independent biological replicates (**h-l,n,o**, *n* = 3). Two-tailed unpaired Student's *t* test (h-l and n) was used for statistical analysis. Two-sided Mann-Whitney U test (**o**) was used for statistical analysis.

of *Bmx* by siRNA reduced the difference in ATP6V1E1 phosphorylation between H37Rv and H37RvΔChp2-infected macrophages (Fig. 6b). Moreover, we detected the ATP6V1E1 phosphorylation in peritoneal macrophages from WT and *Bmx* [+/-] mice infected with H37Rv and H37RvΔChp2. The results showed deletion of Chp2 led to a decrease in ATP6V1E1 phosphorylation and *Bmx* deletion abolished the inhibition of ATP6V1E1 phosphorylation that is induced by deletion of Chp2 (Fig. 6c). This genetic evidence powerfully confirms that Bmx is the primary kinase responsible for phosphorylating ATP6V1E1 in response to Mtb infection. The in vitro phosphorylation assay showed that Chp2 enhanced the BMX mediated phosphorylation of ATP6V1E1 (Fig. 6d). We then examined the role of Chp2 in the assembly of V-ATPase. The results showed that the deletion of Chp2 markedly increased the levels of V1 domain of V-ATPase in membrane fractions (Fig. 6e, f), indicating Rv1184 inhibits the assembly of the V-ATPase at lysosomal membranes. Moreover, the inhibition of *Bmx* by siRNA reduced the difference in intracellular bacterial loads between H37Rv and H37RvΔChp2-infected macrophages (Fig. 6g, h). Treatment with NH4Cl reversed the effect of Chp2 deletion on Mtb survival in macrophages transfected with control siRNA, and while in *Bmx* knockdown macrophages, NH4Cl treatment promoted the growth of both H37RvΔChp2 and H37Rv (Fig. 6g, h), suggesting that Chp2 promotes intracellular survival of Mtb by increasing BMX-mediated phosphorylation of ATP6V1E1 to inhibit lysosomal acidification.

We next investigated the mechanism underlying the promotion of ATP6V1E1 phosphorylation by Chp2. Surface plasmon resonance (SPR) measurements demonstrated that BMX binds ATP6V1E1 with a dissociation constant (Kd) of 190 nM, indicating a high-affinity interaction. (Fig.6i). Given that Chp2 interacts with ATP6V1E1 and promotes its phosphorylation, we next investigated whether Chp2 regulates the binding of the ATP6V1E1 and BMX. Overexpression of Chp2 in HEK293T cells increased the interaction between ATP6V1E1 and BMX (Fig. 6j). Moreover, we observed that the recruitment of endogenous BMX to ATP6V1E1 was reduced in macrophages infected with H37Rv compared to uninfected, and that deletion of Chp2, significantly reduced endogenous interaction of BMX with ATP6V1E1 (Fig. 6k). The immunofluorescence experiments confirmed that Chp2 is present in the host cytoplasm and partially co-localizes with LysoTracker, providing spatial evidence for its translocation into the host cell (supplementary Fig. 5a). Consistently, we also observed that deletion of Chp2 significantly reduced the endogenous phosphorylation level of ATP6V1E1 in macrophages (Fig. 6k), suggesting that Chp2 enhances the binding of ATP6V1E1and BMX to promote the phosphorylation of ATP6V1E1. Using AlphaFold3 structural simulations, we revealed a potential interaction model in which Chp2, BMX, ATP6V1E1, and ATP6V1G1 form a stable complex. In this model, Chp2 acts as a large scaffold protein, promoting the spatial proximity between the catalytic domain of BMX and Tyr[56/57] of ATP6V1E1, thereby facilitating their

interaction and the phosphorylation process (Fig. 6l, supplementary data 1). Taken together, these data suggest that Chp2 acts as a scaffold, facilitating the formation of the Chp2-ATP6V1E1-BMX complex, which enhances the binding of ATP6V1E1 to BMX and increases the phosphorylation levels of ATP6V1E1. This phosphorylation inhibits V-ATPase assembly and suppresses lysosomal acidification, ultimately promoting the intracellular survival of Mtb.

### Inhibition of BMX promotes Mycobacterium clearance

Lysosomal acidification is essential for the Mtb clearance, making lysosome acidification modulators promising targets for host-directed therapy (HDT) against tuberculosis. BMX-IN-1, the first successfully synthesized selective BMX inhibitor[31], was evaluated for its therapeutic effect on the pathogenesis of Mtb. The mice were infected with the Mtb for 14 days, followed by treatment with BMX-IN-1 every two days for an additional 14 days (Fig. 7a). Treatment with BMX-IN-1 led to a significant reduction in bacterial burden and much less pathological damage in the wild-type mice (Fig. 7b-e). However, the therapeutic benefits of BMX-IN-1 were abolished in *Atp6v1e1*[+/-] mice (Fig. 7b-e), indicating that BMX-IN-1 has a therapeutic role for Mtb infection through targeting ATP6V1E1. Moreover, to investigate whether the therapeutic efficacy of BMX inhibitor involves targeting Chp2, we infected mice with H37Rv or H37RvΔChp2. Two weeks post-infection, we administered the inhibitor and assessed pulmonary pathology and bacterial load two weeks after treatment. The data showed that the administration of the BMX inhibitor significantly reduced the bacterial burden and pathological damage in mice infected with H37Rv. However, this therapeutic effect was abolished in mice infected with the H37RvΔChp2 strain(Fig. 7f-i), suggesting that BMX inhibitor play a role via Chp2, indicating that the efficacy of the BMX inhibitor is dependent on the presence of the bacterial effector Rv1184c. It supports a model wherein the inhibitor acts by targeting the host pathway that is subverted by Rv1184c, thus neutralizing the bacterium's virulence strategy. This suggests that host-directed therapy (HDT) with BMX inhibitors could be particularly effective against Mtb strains that rely on this specific mechanism. Collectively, targeting lysosomal acidification by BMX-IN-1 holds potential as a strategy for host-directed therapy in tuberculosis.

## Discussion

Mtb capable of evading lysosomal degradation systems via inhibiting phagosome-lysosome fusion. However, whether Mtb manipulates lysosomal acidification to avoid degradation, remains largely unknown. In this study, we demonstrated that the E1 subunit of V-ATPase was essential for the induction of lysosomal acidification and containment of intracellular bacteria in response to Mtb infection. ATP6V1E1 could be phosphorylated at Tyr[56/57] through tyrosine kinase BMX and the phosphorylation of ATP6V1E1 inhibited the V-ATPase

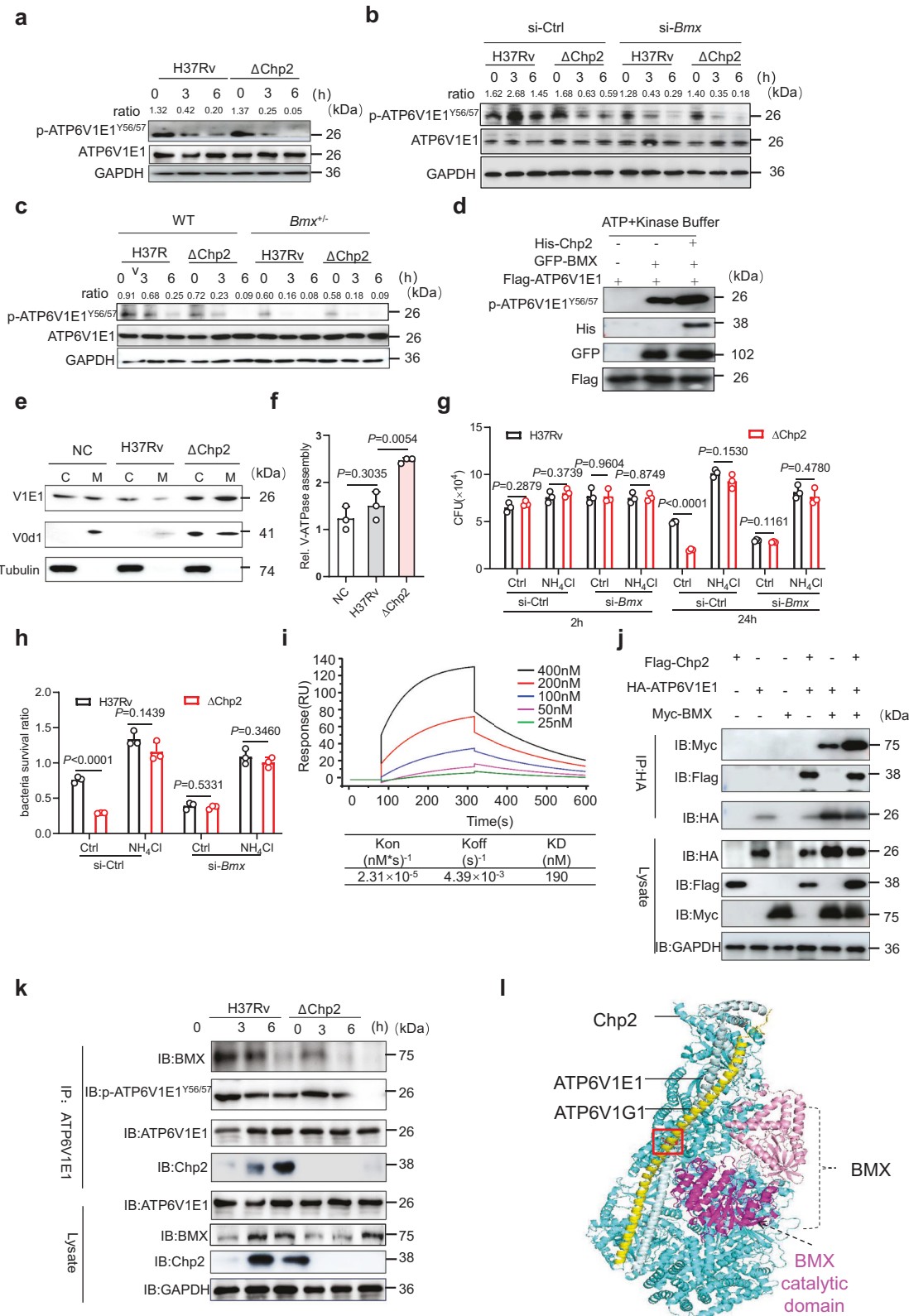

assembly, which in turn suppressed the lysosomal acidification. However, the mycobacterial secreted protein Chp2 promoted BMX-mediated phosphorylation of ATP6V1E1 through enhancing the binding of BMX and ATP6V1E1, thus promoting the intracellular survival of Mtb. Importantly, inhibition of BMX by BMX-IN-1 markedly protected mice from Mtb infection (supplementary Fig. 6). Thus, our findings not only reveal a lysosomal acidification regulatory mechanism mediated

by ATP6V1E1 phosphorylation but also highlight Chp2 as a virulence factor of Mtb manipulates lysosomal acidification.

V-ATPase is an ATP hydrolysis-driven proton pump that plays a critical role in the acidification of phagosomes or lysosomes[15]. The V-ATPase complex consists of two major components: the ATP-hydrolytic V1 regions and the proton-translocation V0 regions. Each region is composed of different subunits[32–35] The functional V-ATPase

**Fig. 6 | Chp2 promotes the binding of ATP6V1E1 and BMX. a** Detection ATP6V1E1 phosphorylation in macrophages infected with H37Rv or H37RvΔChp2 (MOI = 5) for indicated times. ratio: p-ATP6V1E1$^{Y56/57}$/ATP6V1E1. **b** Immunoblotting of the lysates of macrophages transfected with scrambled or *Bmx*-specific siRNA and infected with H37Rv or H37RvΔChp2 (MOI = 5) for the indicated times by p-ATP6V1E1$^{Y56/Y57}$ antibody. ratio: p-ATP6V1E1$^{Y56/57}$/ATP6V1E1. **c** Immunoblot analysis of ATP6V1E1 phosphorylation in peritoneal macrophages isolated from wild-type or *Bmx*$^{+/-}$ mice infected with H37Rv or H37RvΔChp2 for the indicated times (MOI = 5). ratio: p-ATP6V1E1$^{Y56/57}$/ATP6V1E1. **d** In vitro phosphorylation assay of purified GFP-BMX and Flag-ATP6V1E1, which were overexpressed in HEK293T cells and treated with recombinant Chp2 protein(1 μg). **e** Peritoneal macrophages were infected with H37Rv or H37RvΔChp2 (MOI = 5) for 6 h, and changes of V-ATPase subunit abundance in the cytosolic (C) and membrane (M) fractions were analyzed by western blot. **f** Quantification of Western blots described in (**e**) (mean ± SEM). **g,h** Intracellular CFU and bacterial survival in macrophages transfected with scrambled or *Bmx*-specific siRNA and infected with H37Rv or H37RvΔRv1184c for

the indicated times (MOI = 5) or then treat with NH$_4$Cl (20 μM) (mean ± SEM), **g** $P$ = 4.8736E-06 (Ctrl, si-Ctrl, 24 h). **h** $P$ = 4.86583E-05 (Ctrl, si-Ctrl). (**i**) Surface plasmon resonance (SPR) assay of the direct interaction of ATP6V1E1 with BMX. BMX captured on a carboxy (COOH) chip can bind ATP6V1E1 with an affinity constant of 190 nM as determined in a localized surface plasmonic resonance (LSPR) assay. **j** Immunoblotting and immunoprecipitation of lysates from HEK293T cells transfected with plasmids for 48 h were performed and detected using the p-ATP6V1E1$^{Y56/Y57}$ antibody. **k** Endogenous interaction of ATP6V1E1 with BMX in mice peritoneal macrophages infected with H37Rv or H37RvΔRv1184c (MOI = 5) for the indicated times. **l** The predicted structural model of Rv1184c-BMX-V-ATPase complex by AlphaFold3 Server. Yellow represents ATP6V1G1, gray represents ATP6V1E1, blue represents Rv1184c, light purple represents BMX, dark purple represents the catalytic region of BMX, and red box represents the $^{Y56/57}$ site of ATP6V1E1. Data in (**a**–**k**) are representative of one experiment with at least three independent biological replicates (**f**–**h**, $n$ = 3). Two-tailed unpaired Student's *t* test (**f**–**h**) was used for statistical analysis.

complex requires the assembly of subunits in a precise stoichiometry. Mark A. Compton *et al.* reported that Vma9p (subunit e) acts as a rate-limiting factor in V-ATPase assembly in yeast while its overexpression enhances complex formation and boosts proton pump activity[36]. Vma9p is a small hydrophobic protein that is conserved from fungi to animals. In this study, we found that the E1 subunit of V-ATPase exhibited the strongest capacity to regulate lysosomal acidification. We hypothesize that V1E1 overexpression may bypass the normal stoichiometric constraints by acting as a rate-limiting assembly factor, thereby promoting the assembly of more functional complexes and enhancing proton pumping. However, whether and how the subunits of V-ATPases are involved in anti-TB immunity remains unknown. Upon Mtb infection, ATP6V1E1 significantly enhanced lysosomal acidification and inhibited Mtb survival in macrophages and in vivo through targeting V-ATPase. It has been reported that ATP6V1E1 is essential for energy coupling involved in the acidification of acrosomes by regulating proton transport and ATP hydrolysis[37,38]. Our study confirms that ATP6V1E1 acts as a lysosomal acidification modulator and serves as a host factor for the immune defense against Mtb infection.

Recently, it has been reported that the phosphorylation of V-ATPase subunits affects lysosomal acidification. For example, the A subunit of V-ATPase is phosphorylated by protein kinase A (PKA), AMP-activated protein kinase (AMPK), or Aurora kinase A (AURKA)[39,40], while the C subunit of V-ATPase is phosphorylated by WNK (with no lysine (K)) protein kinases[41]. However, the mechanisms by which subunit phosphorylation regulates lysosomal acidification remain unclear. In this study, we found the E1 subunit of V-ATPase was phosphorylated by the tyrosine kinase BMX at tyrosine residues 56 and 57. Phosphorylated ATP6V1E1 inhibited V-ATPase assembly, thereby suppressing lysosomal acidification. Our finds demonstrated a mechanism for the regulation of lysosomal acidification by which BMX-mediated phosphorylation of ATP6V1E1. Future studies, including the generation of ATP6V1E1$^{Y56/Y57A}$ knock-in mice, will be essential to further elucidate the role of ATP6V1E1 phosphorylation in lysosomal acidification and its involvement in other physiological and pathological processes.

The specific mechanism of how phosphorylation of ATP6V1E1 regulates V-ATPase assembly remains unclear. The EM, mass spectrometry and biochemical evidence demonstrate that the three E and G subunits form three EG-heterodimers, which make up the three peripheral stalks that connect the V1 and V0 regions in the eukaryotic V-ATPase complex[42–45]. In our study, structural comparison by AlphaFold3 simulations between the phosphorylated and the unphosphorylated state of ATP6V1E1 showed the distance between Y$^{57}$ in subunit E and R$^{48}$ in subunit G decreased. We speculate that the conformational change in V-ATPase peripheral stalk may play an important role in enzyme structure and regulation of reversible dissociation, which needs further exploration.

Accumulating evidence identified Mtb was the ability to secrete a set of proteins into the cytoplasm of host macrophages and modified phagosome fusion process to achieve immune escape. However, the research on Mtb directly prevented lysosomal acidification to promote survival was limited. Here, we screened 206 mycobacterial proteins and identified Mtb secreted protein Chp2 promoted the intracellular survival of Mtb by inhibiting lysosomal acidification. Chp2 is an acyltransferase that encodes the Chp2 protein. Chp2-mediated transesterification reactions catalyze the conversion of 2,3-diacyltrehaloses (DAT) substrates into penta-acyltrehalose (PAT)[20,46]. DAT and PAT served not only as structural components of the cell envelope but also enhance the ability of Mtb to proliferate and persist within the infected host, thereby promoting its intracellular survival and modulating host immune responses. However, our data demonstrated that Chp2 manipulated BMX mediated-phosphorylation of ATP6V1E1 to inhibit lysosomal acidification, thus enhancing Mtb survival within macrophages and in vivo. Thus, Chp2 appeared to be a virulence factor that allowed Mtb to evade host immune responses by an unexpected escape mechanism, which suppressed lysosome acidification via modulation of V-ATPase posttranslational modification.

Structural simulation analysis revealed that Chp2 functions as a scaffold protein, facilitating the spatial proximity between the catalytic domain of BMX and ATP6V1E1. This interaction results in enhanced phosphorylation of ATP6V1E1 and a reduction in V-ATPase assembly. Here, we identify a function for Chp2 that is independent of its established role as an acyltransferase regulated the synthesis of the cell wall component polyacyltrehalose to inhibit phagosomal acidification in Mtb. Our study identifies that Chp2 can modulate vesicle acidification through multiple mechanisms, wherein Chp2 can directly interact with host proteins or exert its effects through its catalytic products.

It is well known that V-ATPase-mediated lysosomal acidification is necessary for their biological function. Therefore, targeting V-ATPase-mediated acidification represents a valid strategy for anti-tuberculosis therapy. In our study, we identified BMX as a novel kinase regulating V-ATPase-mediated lysosomal acidification. BMX-IN-1, a potent and irreversible BMX kinase inhibitor[47], effectively reduced the bacterial burden of the Mtb strains in the lung tissue of infected mice. Furthermore, PLGA-aNPs[48], acidic nanoparticles designed to restore defective lysosomal acidification, also decreased the bacterial burden in the lungs of Mtb-infected mice. Consequently, targeting lysosomal acidification appears to be a potential host-directed therapy against TB. Abnormal lysosomal acidification is closely associated with various diseases, including lysosomal storage disorders, neurodegenerative diseases, and cancer. In these contexts, the regulation of lysosomal acidification through ATP6V1E1 phosphorylation remains to be fully explored. Thus, the potential development of drugs that modify

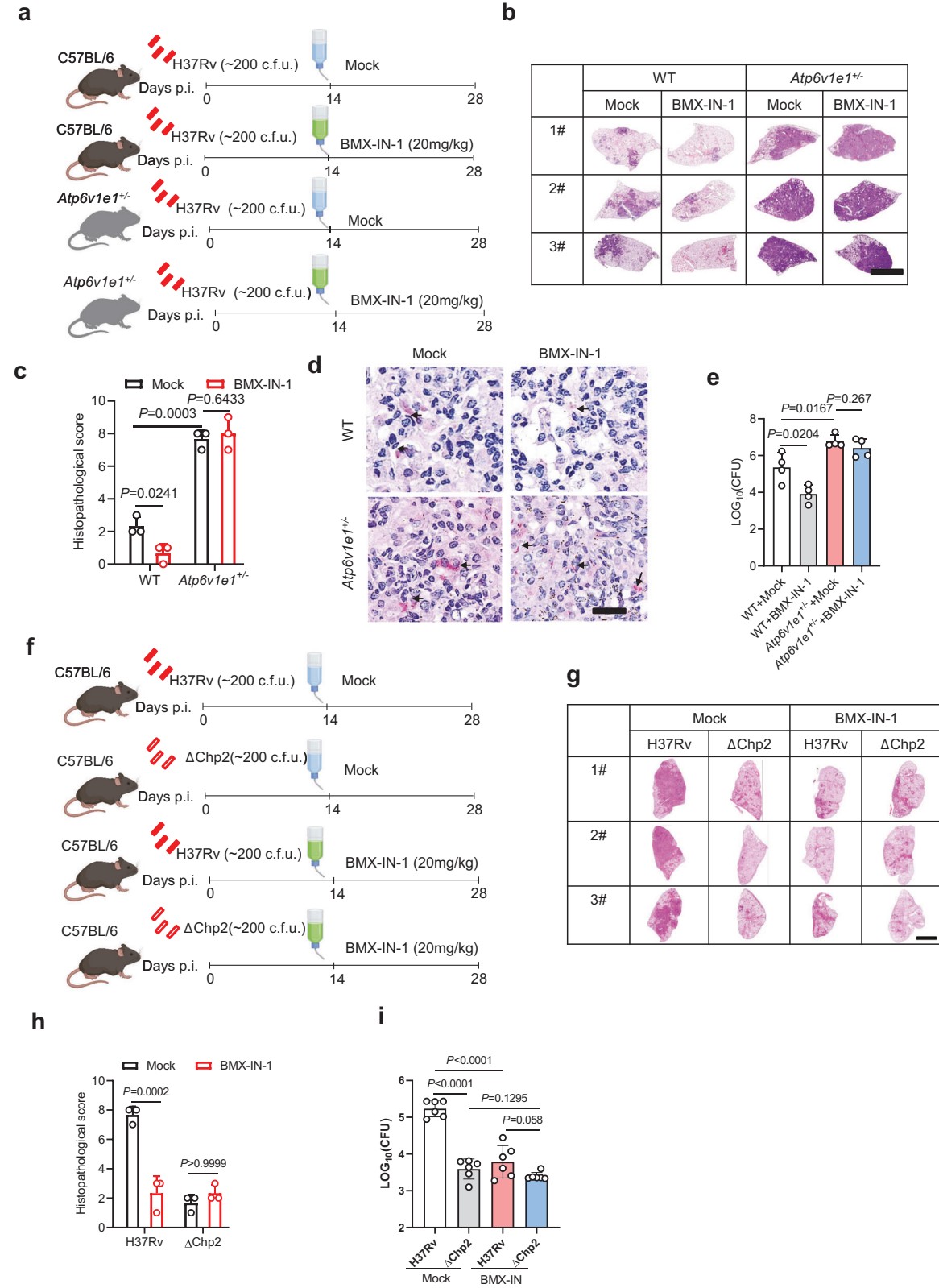

ATP6V1E1 phosphorylation, such as BMX inhibitors, as new therapeutic strategies for these diseases warrants further investigation.

Overall, our study demonstrated not only a crucial role of phosphorylation modification in the regulation of lysosomal acidification, but also an unprecedented immune escape strategy employed by Mtb via manipulating the phosphorylation of ATP6V1E1.

## Methods

### Mice

C57BL/6 mice were purchased from the Shanghai Laboratory Animal Center (Shanghai, China) and maintained under specific pathogen free (SPF) conditions at the Laboratory Animal Center of Tongji University. Female mice aged 6-8 weeks were used for all experiments. *Atp6v1e1*[+/-] mice and *Bmx*[+/-] mice were generated using the CRISPR/Cas9 method[49].

**Fig. 7 | Inhibition of BMX promotes Mycobacterium clearance. a** Schematic diagram of the BMX-IN-1therapeutic effect on wild-type and *Atp6v1e1*[+/-] mice. **b** H&E staining of the lung tissues of WT and *Atp6v1e1*[+/-] mice infected with H37Rv by intranasal infection ( ∼ 200 CFU) for 2 weeks and oral administration of BMX-IN-1 (20 mg/kg) for another 2 weeks. 1-3# indicate three representative lung tissues. Scale bar, 5000 μm. (**c**) Quantification of the histopathology score in the lungs of mice infected with H37Rv as in (**b**) (mean ± SEM). **d** Acid-fast staining of the lung tissues of mice infected with H37Rv as in (**a**). Scale bar, 500 μm. **e** Bacterial CFU of the lung tissues of mice infected with H37Rv as in (**a**) (mean ± SEM, *N* = 4 mice). **f** Schematic diagram of the BMX-IN-1therapeutic effect on mice infected with the indicated strains. **g** Representative H&E staining of the lung tissues of mice infected

with H37Rv or H37RvΔChp2 by intranasal infection ( ∼ 200 CFU) for 2 weeks and oral administration of BMX-IN-1 (20 mg/kg) for another 2 weeks. 1-3# indicate three representative lung tissues. Scale bar, 5000 μm. **h** Quantification of the histopathology score in the lungs of mice infected with H37Rv or H37RvΔChp2 as in (**f**) (mean ± SEM). **i** Bacterial CFU of the lung tissues of mice infected with H37Rv or H37RvΔChp2 as in (**f**) (mean ± SEM, *N* = 6 mice). *P* = 4.80346E-07 (Mock), *P* = 3.03917E-05 (H37Rv). Data in (**a**–**i**) are representative of one experiment with at least three independent biological replicates (**c,h**, *n* = 3). Two-tailed unpaired Student's *t* test (c and h) was used for statistical analysis. Two-sided Mann-Whitney U test (e and i) was used for statistical analysis. Elements in Fig. 7 (**a,f**) were created in BioRender[50].

Briefly, in vitro-translated Cas9 mRNA and guide RNAs (gRNAs) were co-microinjected into fertilized C57BL/6 zygotes. The sequences of gRNA were listed in Supplementary Table 1. Frameshift mutations in the founders were screened by T7E1 assay and confirmed by DNA sequencing. Three F1 founders were obtained for each line, and a single founder from each was selected for backcrossing to C57BL/6 mice to establish the strain. *Atp6v1e1*[+/-] mice and *Bmx*[+/-] mice were identified by PCR using genomic DNA isolated from tail tissue. The sequences of PCR primes were listed in Supplementary Table 2.

## Plasmids, antibodies and reagents

Plasmids used in this study were described in Supplementary Table 3. Antibodies against phosphorylated V1E1[Y56/57], total anti-V1E1 and anti-Chp2 were produced by ABclonal Technology (Wuhan, China). Antibodies against BMX 24733S) and GAPDH (2118S) were obtained from Cell Signaling Technology (Beverly, MA) and used at 1:1000 for Western blotting. The anti-phosphotyrosine monoclonal antibody (4G10) (05-321) was obtained from Merck. Anti-hemagglutinin (HA, H6908) and mouse monoclonal anti-Flag(F3165) from Sigma-Aldrich. Rabbit monoclonal anti-His (AE086), rabbit monoclonal anti-Myc (AE070), goat anti-rabbit IgG secondary antibody (AS014) and goat anti-mouse IgG secondary antibody (AS003) were purchased from ABclonal. Anti-V0d1 antibody(18274-1-AP), anti-tubulin antibody (80762-1-RR), anti-Na-K antibody (13884-1-AP) were purchased from Proteintech. TSA 7-color kit (abs50015-100T) was purchased from Absinbio. Antibodies against CD3 (78588) and CD20 (70168 T) were purchased from Cell Signaling Technology (Beverly, MA). Antibodies against MPO (ab208670) and CD11b (ab133357) were purchased from Abcam. Anti-SigA (663205) were purchased from BioLegend. Antibody against LAMP1 (S-952-18) was purchased from STARTER. Fluorescent secondary antibodies were purchased from ThermoFisher and used according to the manufacturers' instructions. FITC-dextran (A642107) was purchased from Sangon Biotech. RPMI 1640 medium(C11875500CP), DMEM(SH30249.01), fetal bovine serum (FBS) (26170043), penicillin and streptomycin(15140163) were purchased from Invitrogen (Shanghai, China). LysoTracker™ Red DND-99 (L7528) and pHrodo™ Green Dextran (P35368) were obtained from Invitrogen™. The Magic Red™ Cathepsin B Assay Kit (ICT-937) was purchased from Enzo Life Sciences. The shRNA library of non-receptor tyrosine kinases was originally from the MISSION® LentiPlex® Human Pooled shRNA Library TRC2 from Sigma Aldrich. The siRNA targeting mouse *Bmx* (sc-38942) and control scrambled siRNA (sc-37007) were obtained from Santa Cruz Biotechnology, and a siRNA pool specifically targeting mouse *Atp6v1e1* was obtained from RiboBio (Guangzhou, China). Bafilomycin A1(Baf-A1, HY-100558) was purchased from Med Chem Express. BMX-IN-1(1431525-23-3) was purchased from ApexBio.

## Bacterial culture and infection

*Mycobacterium tuberculosis* H37Rv strain was cultured to mid-log phase in Middlebrook 7H9 broth (Becton Dickinson) supplemented with 0.05% Tween-80 and 10% oleic acid-albumin-dextrose-catalase (OADC, 211886, Becton Dickinson, Sparks, MD).

H37RvΔChp2 was constructed by Shanghai Gene-optimal Science & Technology Co., Ltd. The shuttle vector pMV261 (provided by K. Mi, Institute of Microbiology, Beijing, China) was used to complement the strain H37RvΔChp2 with wild-type Chp2. Expression of Chp2 in mycobacteria was examined by immunoblot analysis. For H37RvΔChp2, 50 μg/mL hygromycin B was added to culture. For H37Rv (ΔChp2+ Chp2), 50 μg/mL hygromycin B and kanamycin were added to culture.

For macrophages infection, mice peritoneal macrophages were seeded in 12-well plates, and cultured for 24 h at 37 °C in a 5% CO2 incubator. Mtb strains were added to cells at a multiplicity of infection (MOI) of 5.

## FITC labeling of Mtb

Mid-log phase Mtb cultures were harvested by centrifugation, washed twice, and resuspended in pre-chilled 0.1 M carbonate-bicarbonate buffer (pH 9.0). FITC isomer I was added to a final concentration of 0.1 mg/mL, and the suspension was incubated at 37 °C for 1 hour in the dark with gentle agitation. After labeling, bacteria were washed thoroughly 4-5 times with pre-chilled phosphate-buffered saline (PBS, pH 7.4) until the supernatant was fluorescence-free to remove unbound dye. Finally, the labeled bacteria were resuspended in appropriate infection medium and quantified using McFarland standards coupled with colony-forming unit (CFU) plating. All labeling and infection steps were performed under light-protected conditions. In the image analysis phase, fields containing visible large clumps ( > 5 μm in diameter or containing clearly fused bacterial signals) were systematically excluded from quantification.

## Cells culture

HEK293T cells (ATCC CRL-3216) were obtained from the American type culture collection (ATCC). The cells were maintained in Dulbecco's modified Eagle's medium (DMEM, Hyclone) supplemented with 10% (v/v) heated-inactivated fetal bovine serum (FBS, Gibco) and 100 U/ml penicillin-streptomycin. Routine testing was conducted to ensure the cells were free of mycoplasma contamination.

## Mice macrophages isolation

Peritoneal macrophages were isolated as previously described. Briefly, mice were injected intraperitoneally (IP) with 2.0 ml of 4% Brewer's thioglycollate medium (Sigma–Aldrich). Three days later, primary macrophages were harvested by peritoneal lavage with 10 mL cold PBS from euthanized animals. Subsequently, the cells were plated in 12-well plates at 10[6] cells per-well in RPMI 1640 supplemented with 10% FBS and penicillin-streptomycin, and then incubated at 37 °C in 5% CO2 for 2 h. The cultures were then washed three times with PBS to remove nonadherent cells, and the remaining adherent monolayer cells were used as primary peritoneal macrophages. The bone marrow-derived monocytes (BMDMs) were generated by isolating mouse bone marrow cells and culturing them for 7 days in medium supplemented with 10% (v/v) FBS, 40 ng/ml M-CSF (Peprotech), and 20 ng/ml IL-4 (Peprotech).

## Cell fractionation

To measure the V-ATPase assembly, the cells fractionation was performed to obtain cytosolic and membrane fractions. HEK293T cells were transfected with plasmids encoding HA-ATP6V1E1, HA-ATP6V1E1$^{Y56/57A}$ and HA-ATP6V1E1$^{Y56/57E}$ for 48 h, or primary peritoneal macrophages from wild-type and $Bmx^{+/-}$ mice pretreated with BMX-IN or not were prepared. Monolayers were rinsed with ice-cold PBS and scraped into 500 μL ice-cold fractionation buffer (250 mM sucrose, 10 mM HEPES, pH 7.2, 1 mM EDTA, 1 mM PMSF, 5 μg/mL leupeptin, 2 μg/mL aprotinin, 1 μg/mL pepstatin, 2 mM β-glycerophosphate, 1 mM NaF). Cell suspensions were centrifuged at $300 \times g$ for 5 min at 4 °C, and the pellets were resuspended in 500 μL fresh ice-cold fractionation buffer. Cells were lysed using a cell homogenizer, and crude lysates were cleared of unbroken cells and nuclei by centrifugation at $1000 \times g$ for 10 min at 4 °C. The resulting supernatants were centrifuged at $100{,}000 \times g$ for 30 min at 4 °C in the ultracentrifuge to pellet membrane fractions. The supernatants (cytosolic fractions) were concentrated, and membrane pellets were solubilized in fractionation buffer supplemented with 1% SDS.

## Immunoblotting and immunoprecipitation

For immunoblotting analysis, the cells were lysed in RIPA buffer (10 mM Tris-Cl, pH 8.0, 150 mM NaCl, 1% Triton X-100, 1% Na-deoxycholate, 1 mM EDTA, 0.05% SDS) supplemented with freshly added protease inhibitors. Protein concentration was determined by the Bradford assay (Bio-Rad). Equal amounts of protein were separated by SDS–PAGE and transferred to nitrocellulose membranes (Millipore). Membranes were blocked with 5% bovine serum albumin and incubated with primary antibodies at 4 °C overnight, followed by HRP-conjugated secondary antibodies Signals were detected with ECL (GE Healthcare) or SuperSignal West Dura (Thermo Scientific) and visualized on autoradiographic film.

For immunoprecipitation (IP), cells were lysed in Western blot and IP cell lysate buffer supplemented with protease inhibitor cocktail (MedChemExpress). Following lysis, the samples were subjected to centrifugation, and the resulting supernatants were incubated overnight at 4 °C with either anti-FLAG M2 Magnetic Beads or HA Beads (Sigma-Aldrich). The samples were washed three times with PBST (KH2PO4 2 mM, Na2HPO4 8 mM, NaCl 136 mM, KCL 2.6 mM, 1% Triton X-100) and subjected to Western blot analysis.

## Immunofluorescence staining and confocal microscopy analysis

HEK293T cells were grown in glass bottom cell culture dish (NEST-SCIENTIFIC 801002). Adherent primary peritoneal macrophages were infected with Mtb at MOI of 5 for 4 h, fixed in 4% formaldehyde overnight, and then washed twice with PBS for 5 min. Cells were blocked with 1% BSA for 30 min at room temperature and then incubated with a primary antibody (1:200) overnight at 4 °C. Subsequently, samples were incubated with a secondary antibody (1:400) for 2 h at 37 °C. Nuclei were counterstained with 4,6-diamidino-2-phenylindole (DAPI). Images were acquired on a Leica SP5 confocal microscope using a 63× objective.

For all microscopy-based LysoTracker experiments, fluorescence intensity was quantified in a minimum of 100 individual cells from multiple randomly selected fields of view. This analysis was repeated across three independent biological replicates. The sample size (n) stated in the figure legends refers to the number of biological replicates (each containing data from >100 cells).

## High-throughput fluorescence screening and analysis

HEK293T cells were seeded onto 96-well microplates and allowed to attach overnight. The next day, cells were transfected with plasmids encoding Mtb-secreted proteins or lipoproteins for 48 h. A miniaturized fluorescence-based Lyso-tracker was performed in 96-well microplates with a final volume of 50 μL per well to assess lysosomal acidification. After the staining, fluorescence (Ex/Em = 577/590 nm)

was measured on a Thermo Scientific™ ArrayScan™ multimode microplate reader using top-read optics; instrument sensitivity was fixed at a pre-optimized gain to ensure consistency across the screen. Raw fluorescence units (RFU) were normalized per plate to percent inhibition relative to the median uninhibited enzyme control wells (maximum signal) and median no-enzyme control wells (background), which were included in designated columns on each plate. Normalized data were processed and analyzed for hit identification.

## Multiplex Immunohistochemistry (mIHC)

Multiplexed immunohistochemistry (mIHC) was performed by staining 4-um-thick formalin-fixed, paraffin-embedded whole tissue sections with standard, primary antibodies sequentially and paired with Five Color Multiplex Fluorescent lmmunostaining Kit (abs50013, Absinbio, Shanghai). Then by staining with DAPI. The deparaffinized slides from lung tissues of mice were incubated with anti-CD3 antibody (CST 78588), for 30 minutes and then treated with anti-rabbit/mouse horseradish peroxidase-conjugated (HRP) secondary antibody(abs50015-02, Absinbio, Shanghai) for 10 min. Then labelling was developed for a strictly observed 10 minutes, using TSA 570 per manufacturer's direction. Slides were washed in TBST buffer and then transferred to preheated citrate solution (90 °C) before being heat-treated using a microwave set at 20% of maximum power for 15 min. Slides were cooled in the same solution to room temperature. Between all steps, the slides were washed with Tris buffer. The same process was repeated for the following antibodies/fluorescent dyes, in order: anti-MPO (ab208670)/ TSA 620, anti-CD11b (ab133357)/ TSA 700, anti-CD20 (70168 T)/TSA 520. Each slide was then treated with 2 drops of DAPI, washed in distilled water, and manually cover slipped. Slides were air dried, and take pictures with Pannoramic MIDI II (3DHIS-TECH). Images were analyzed using Indica Halo software.

## Lysosomal content assay

To assess lysosomal acidification, primary peritoneal macrophages were infected with Mtb at MOI of 5 for 4 h and then stained with two fluorescent dextrans: Oregon Green™ 488- conjugated Dextran (10,000 MW, Anionic, quenched in acidic environments), and Tetramethylrhodamine- conjugated Dextran (10,000 MW, Anionic, Lysine Fixable, pH-independent) (Thermo Fisher). After the addition of these two dextrans, cells were incubated in a glass-bottom dish (NEST) at 37 °C for 6 h, the medium was replaced, and incubation was continued overnight. Labeled cells were then washed into HBSS, and imaged by fluorescence microscope. Green and red fluorescence intensities were quantified using a BZ-X800 (KEYENCE).

As an alternative approach, LysoTracker Red DND-99 was used to evaluate lysosomal acidification. Primary peritoneal macrophages plated in glass-bottom culture dishes (NEST) were infected with Mtb at MOI of 5 for 4 h, and then treated with 25 μM monensin for 20 minutes, incubated with1 μM LysoTracker for 5 min, washed twice with PBS and nuclei were stained with DAPI. Images were captured by confocal microscopy (Leica TCS SP8).

To generate a lysosomal pH calibration curve, primary peritoneal macrophages were equilibrated in calibration buffer of varying pH (5 mM NaCl, 115 mM KCl, 1.2 mM MgSO$_4$, 25 mM MES; pH 3.5–7.5). Samples were excited with 357–373 nm and emission were collected in the blue (W1; 417–483 nm) and yellow (W2; 490–530 nm) channels. Ratiometric (W1/W2) images were computed using the Image Calculator function in MetaMorph (Molecular Devices).

## RNA interference and transfection

Mice peritoneal macrophages were transfected with small interfering RNAs using the transfection reagent (AM4511, Thermo Fisher) according to the manufacturer's instructions. The individual siRNAs targeting mouse $Bmx$, $Atp6v1e1$ and a scrambled siRNA were each diluted in RNase-free water to final volume of 330 μL.

## Intracellular CFU assay

Adherent primary murine peritoneal macrophages were infected with the H37Rv strain or other H37Rv strains at a multiplicity of infection (MOI) of 5 for 2 hours at 37 °C. Following infection, the cells were washed twice with phosphate-buffered saline (PBS). To determine the number of viable intracellular bacteria, the infected cells were lysed at 2 h or 24 h post-infection using 0.1% (v/v) Triton X-100 (ThermoFisher, Cat#85111) in PBS. The lysates were then serially diluted, plated on agar, and the bacterial burden was quantified by counting the resulting colony-forming units (CFU). The BMDMs were infected with the H37Rv strain or H37RvΔChp2 for 2 h, 24 h, and 48 h to detect CFU.

## Generation of phospho-ATP6V1E1$^{Y56/57}$ antibodies

A rabbit polyclonal antibody recognizing ATP6V1E1 phosphorylated at Y56/57 (designated p-ATP6V1E1$^{Y56/57}$) was generated in collaboration with ABclonal Biotech. Two rabbits were immunized with a 1:1 mixture of the phosphopeptides c(KLH)52KIMEY(p-Y) Y(p-Y) EK59 and c(KLH) 54MEY(p-Y) Y(p-Y) EKKE61, (KLH conjugation via an N-terminal cysteine; p-Y denotes phosphotyrosine). The corresponding non-phosphorylated peptide c(KLH)-KIMEYYEKKE was used for affinity purification and antibody validation.

## In vitro kinase assay

Recombinant BMX (1 μg) was incubated with recombinant ATP6V1E1 (1 μg) in 3 μL of 10× phosphorylation buffer (200 mM Tris-Cl, pH 7.5, 500 mM KCl, 100 mM MgCl$_2$), 1 μL 2 pM of ATP, and complete protease inhibitor mixture (Roche). The reaction was brought to a final volume of 30 μL with distilled water and incubated for 45 min at 30 °C. Then, the reaction mixture was subjected to SDS-PAGE.

## Intranasal infection (i.n.) of mice with Mtb

Mice were infected intranasally with 4×10$^5$ viable CFUs of H37Rv, H37RvΔChp2 or H37Rv (ΔChp2 + Chp2). The inoculum (40 μL per mouse) was applied to the nares, alternating between the left and right nostrils, in a Biosafety Level-3 (BSL-3) laboratory. Mice were sacrificed 28 days post-infection, and bacterial burden in the lungs was determined by plating serial tenfold dilutions of individual lung homogenates on Middlebrook 7H10 agar supplemented with 10% OADC. Mtb colonies were incubated at 37 °C and counted after 21 days. For BMX-IN-1 intervention, after 2 weeks of infection, mouse was given a dose of 20 mg/kg BMX-IN-1 working solution through oral administration for 2 weeks. Four weeks after infection, the mice were sacrificed, and the bacterial loads in the lungs were determined by CFU counts. To assess the effect of Baf-A1, mice were intraperitoneally injected with Baf-A1 prior to infection for 3 days, followed by the weekly Baf-A1(10 mg/kg, 200 uL/mice) injections for 4 weeks. Finally, the mice were sacrificed, and CFU counts in the lungs were measured to determine the bacterial load.

## Chp2 protein administration

To assess the direct pathogenic potential of the Chp2 protein, C57BL/6 mice were injected with recombinant Chp2 protein via either the intravenous (100 μg/mL, 200 μL per mouse) or intranasal (100 μg/mL, 30 μL per mouse) route. After three weeks, histopathological damage in the lungs from both treatment groups was assessed by H&E staining.

## Histological analysis and acid-fast staining

Lung tissues from infected mice were fixed in 4% phosphate-buffered formalin for 24 h, then embedded in paraffin. Serial sections were cut from the tissue blocks, and five non-continuous sections were selected for Hematoxylin and eosin (H&E) staining to assess inflammatory cell infiltration. The stained slides were examined by light microscopy, and histopathological evaluation was performed by pathologists blinded to group identity. The pathology scores (0-10) were based on the ratio of inflammatory cell infiltration area to the total lung section area on the maximal cross-section of a lung lobe, with heavily stained areas indicating infiltration and damage, and lighter areas representing normal alveolar structure. Acid-fast staining (Ziehl–Neelsen) was performed to detect mycobacteria.

## Surface plasmon resonance (SPR)

Interaction between ATP6V1E1 and BMX or Chp2 was analyzed using OpenSPR™ (Nicoya Lifesciences, Waterloo, Canada). A carboxyl-functionalized sensor chip (COOH chip) was installed, and the instrument was primed with PBST (pH 7.4) at a maximum flow rate of 150 μL/min until the baseline signal stabilized. The sample loop was then rinsed and emptied. The flow rate was reduced to 30 μL/min prior to chip activation with a freshly prepared 1:1 EDC/NHS solution. Ligand (200 μL) diluted in activation buffer was injected for 4 min. After ligand immobilization and buffer rinses, the surface was blocked with 200 μL blocking solution, rinsed and dried with air. Baseline stability was confirmed for 5 min. The flow rate was returned to 150 μL/min and a regeneration solution was used as required to remove residual analyte. Analytes were prepared at the concentrations indicated in the results and injected at 30 μL/min. Association and dissociation phases were 120 s and 300 s, respectively. Sensorgrams were fitted using Trace-Drawer (Ridgeview Instruments AB, Sweden) with a one-to-one binding model.

## AlphaFold3 structural simulations

The human V-ATPase structure (PDB: 7U4T), previously published, was utilized to analyze the structure of V1E1. Structural simulations were conducted using the AlphaFold3 online server (https://golgi.sandbox.google.com/), and the results were visualized with PyMOL software. The model is available in ModelArchive at https://www.modelarchive.org/doi/10.5452/ma-3dlmq.

## Statistical analysis

Data are presented as mean ± SEM. Differences between two groups were assessed by unpaired two-tailed $t$ tests. For comparisons involving more than two groups, Two-sided Mann-Whitney U test was performed with correction for multiple comparisons where appropriate. All statistical analyses were conducted in GraphPad Prism 8.0. A $P$ value < 0.05 was considered statistically significant.

## Ethics consideration

All animal experiments were conducted in accordance with the National Institutes of Health Guidelines for the Care and Use of Laboratory Animals and were approved by the Animal Experiment Administration Committee of Tongji University School of Medicine (Approval No. TJAA06520101).

## Reporting summary

Further information on research design is available in the Nature Portfolio Reporting Summary linked to this article.

## Data availability

The data are available within the Article, Supplementary Information or Source Data file. Source data are available in the figshare repository. [https://doi.org/10.6084/m9.figshare.30563891].

## Code availability

In this study, no new codes were generated, as all analyses were performed using existing tools and methodologies.

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

## Acknowledgements

We are deeply grateful to all the study participants, research staff, and students who took part in this work. This work was supported by National Key R&D Program of China (2023YFC2307002 to H. L.; 2021YFA1300902 to R. Z.); National Natural Science Foundation of China (No. 82570005 and 82170010 to R. Z.); Tongji University Medicine-X Interdisciplinary Research Initiative (No.2025-0554-YB-17 to J.C.), Key Laboratory of Pathogen-Host Interaction (Tongji University), Ministry of Education (No. KFKT20250002 to J.C.).

## Author contributions

R.Z., B.G., F.L., H.L., and L.L. designed this study. J.C., F.T., L.Q., X.W., H.L., Y.D., F.W., and C.P. performed all experiments and analyzed data, assisted by Z.L., J.W., X.H., L.W., H.Y., L.W., and W.S. Mice infection was supported by X.C. All the *M. tuberculosis* and K.O strains were conserved and sub cultured by X.H. and J.W. All authors discussed the results and commented on the manuscript.

## Competing interests

The authors declare no competing interests.
