## [Transparent Peer Review file · Nature Communications]

Mycobacterium tuberculosis modulates phosphorylation of host ATP6V1E1 to promote intracellular survival

Corresponding Author: Dr Ruijuan Zheng

Version 0:

Reviewer comments:

Reviewer #1

(Remarks to the Author)

In this article, the authors explore how *Mycobacterium tuberculosis* (Mtb) manipulates macrophages to inhibit lysosomal acidification and promote its own replication. They show that Mtb secretes the effector Rv1184c, which targets the V-ATPase acidification machinery, specifically the V1 subunit. A wide range of techniques was used, and substantial experimental data were generated to support their hypothesis. The study is well-constructed and very interesting. Mechanisms by which Mtb blocks lysosomal acidification remain poorly understood, making this work valuable. Furthermore, the *in vivo* inhibition of BMX appears to be a promising host-directed therapy (HDT) strategy. Below are some suggestions to improve the quality of the manuscript:

- In the first section of the results, the authors conclude that the use of bafilomycin to inhibit V-ATPase demonstrates that Rv1184c inhibits lysosomal acidification via this pathway. This conclusion seems a bit premature and could benefit from an additional experimental approach to confirm the hypothesis, even if later experiments do support it further.
- Line 129: Infection in macrophages was monitored for 24 hours. This time point may be too early—what happens at later time points, such as 3 or 4 days post-infection?
- Line 131: Please explain the role of the NH₄Cl control.
- The results show increased acidification during infection with the Rv1184c-deficient mutant strain, and that this strain is attenuated in macrophages. Is it possible to provide imaging data showing co-localization between bacteria and LysoTracker under these conditions? For instance, is there increased co-localization of the Rv1184c deletion mutant with LysoTracker compared to wild-type Mtb?
- Line 135: Could you comment on the reduced immune cell infiltration? Which immune cell types are affected? Could this be assessed by flow cytometry?
- Line 145: Again, I am not fully convinced by the bafilomycin experiment. Inhibiting V-ATPase will necessarily favor infection and may mask the effect of the mutation. Could you consider an alternative readout?
- Is it possible to visualize or measure lysosomal acidification in alveolar macrophages from infected mice?
- Line 180: The *in vivo* results are interesting. Have you considered infecting Atp6v1E1 knockout mice with the Rv1184c mutant strain? It would be interesting to discuss the expected results.
- BMX inhibition shows very promising data. Could this therapy be combined with standard antibiotic treatment to test for potential synergy?
- What would be the effect of BMX inhibition in mice infected with the Rv1184c mutant strain?
- Similarly, what would be the effect of PLGA-aNP treatment in Atp6v1E1 knockout

Reviewer #2

(Remarks to the Author)

This manuscript by Chen et al, titled “*Mycobacterium tuberculosis* manipulates phosphorylation of ATP6V1E1 to promote its intracellular survival”, shows an important phosphorylation-dependent regulation of lysosomal acidification during Mtb infection, with the potential to regulate Mtb survival inside the infected macrophages. The authors show that Mtb secreted protein Rv1184c facilitates interaction between tyrosine kinase BMX and the vATPase subunit ATP6v1e1, leading to increased phosphorylation of ATP6v1e1 and corresponding decline in the lysosomal acidification, resulting in less efficient bacterial killing. In view of that, inhibiting BMX kinase in the animal models shows a decrease in lung CFU burden, which is dependent on the protein ATP6v1e1.

The regulation of lysosomal activity by Mtb during infection has been under immense focus for investigation due to its direct role in killing intracellular bacterial infections. The current manuscript provides interesting insights into this key battle between the host and the pathogen. The authors generate useful models to address the specific questions asked in this study.

However, the study suffers from some serious limitations. There are some general observations and specific concerns pointed out below.

General observations: There is a lack of details on the experiments and results. For studies on Mtb infection, cell/cell-lines used, the time points post-infection, duration of treatment, etc., are vital details. These details are missing even in the methods section. Some of the results reflect a less robust experimental strategy, leading to deviations from the expected line. Different experiments are done on different cell lines/types. The rationale for those is not obvious. Cells used for the experiments are not explicitly mentioned.

Major limitations:

- 1) The ability of Mtb to block phagosome maturation and lysosomal acidification is well-documented. However, throughout this study, a significantly higher number of bacilli is seen in acidified lysosomes, which, of course, increases upon loss of Rv1184c. The observed phenotype is difficult to interpret due to the absence of any staining for the Mtb, which would allow assessing bacterial presence in acidified lysosomes.
- 2) Conclusions for a few results and interpretations made thereof are not aligned with the experimental observations.
- 3) The phenotype of ATP6v1e1 knockout mice is not well explained. With a loss of lysosomal function, in these animals, even without infection, they should show lysosomal storage disorder phenotype.

Specific points:

- How ATP6v1e1 overexpression alone is able to cause significant lysosomal acidification in 293T cells? The vacuolar ATPase is a multi-subunit protein, and the catalytic site, formed by the V1 domain, consists of all subunits. This needs more clarity. What happens to ATP6ve1 expression upon Mtb infection, and whether that differs from the expression of other subunits?
- For lysosomal acidification during Mtb infection, there is no staining for Mtb (Fig. 1g, 3q, fig. S3g, etc.). The virulent Mtb is known to block lysosomal acidification through PMA. Since MOI used is 5, it is important to show that the cells analysed/being shown are also Mtb- infected.
- Why should H37Rv CFU decline in WT cells from 0 hour to 24 hours post-infection? This is a critical observation, Mtb CFU is generally stable or grows during 24 hours, however a decline in CFU indicates loss of virulence. Have the authors checked PDIM levels of the strains used? The observed dip in Mtb CFU in the WT cells needs to be analysed more robustly.
- Tissue sections from uninfected ATP6v1e1 should also be shown. With the loss of lysosomal function, severe pathology may develop due to lysosomal storage disorder like condition.
- Line 178: LysoTracker staining shown in WT H37Rv infection is too high. H37Rv blocks lysosomal acidification. Secondly, fig. 3q-r: the LysoTracker staining upon H37Rv infection visible in the images at 0 hours and 4 hours in the WT and ATP6v1e1^{-/-} cells (3q) is not reflected in the corresponding quantification plot (3r). This is very concerning.
- Fig 6a: Rv1184c deletion: the impact on ATP6v1e1 phosphorylation is not very distinct from the WT infection.
- The Rv1184c knockout and complementation data is clean. However, authors need to show that sufficient Rv1184c gets secreted during infection in macrophages to be able to interact with and alter ATP6v1e1 phosphorylation. So far all such evidences are shown using overexpression experiments but whether expression and secretion of native protein is sufficient to alter ATP6v1e1 phosphorylation and lysosome acidification?
- siBMX treatment: How many hours after BMX knockdown infection was given? What does 0 hour mean here? What is the knockdown efficiency of BMX siRNA? It is strange to see a complete loss of ATP6v1e1 phosphorylation.
- For all phosphorylation immunoblots, GAPDH is not a correct loading control. The control should be the ATP6v1e1 protein itself.
- The PLGA-aNP experiment is not relevant to the current study unless it is shown in combination with the BMX inhibitor.

Minor points:

Typo in line 233: "numerous pathological pathways"?

Authors should use Grammarly or similar software for written English.

Reviewer #3

(Remarks to the Author)

This manuscript by Chen J et al. uncovers a mechanism by which Mycobacterium tuberculosis H37Rv (Mtb) manipulates phagolysosomal maturation to avoid killing by macrophages. The authors show that Mtb protein Rv1184 promotes BMX interaction with the v-H⁺-ATPase subunit ATP6V1E1 thereby enhancing phosphorylation of ATP6V1E1 at tyrosine residues. This inhibits the v-ATPase assembly at the membrane, dampening phagolysosomal acidification and allowing Mtb survival. This manuscript therefore highlights an additional mechanism by which Mtb manipulate phagosome maturation. In its large part, I found this MS very interesting and convincing. It provides many details in the mechanism described with very neat experiments. My enthusiasm was somehow dampened by some flows in the logic that brought the authors to realized some experiments. For example, what brought the authors to assess phosphorylation fo ATP6V1E1 in the first place? Furthermore, there is a duplication of images in Fig 7b and 7g: 1) Mock controls are the same images but it is not mentioned in the legend ; 2) BMX-IN-1 #3 is the same image as PLGA-aNP #2. I must say that my expertise in histological HandE staining in the context of Mtb infection is limited. Thus I could not really evaluate this part of the work.

Some suggestions/comments that may improve the study:

- 1) Fig 1b, do the authors found CISH and PtpA in their screen?
- 2) What is the effect of Rv1184c by itself upon injection in mice?
- 3) In Fig 3B, I understand that the authors over-expressed FLAG-Rv1184c together with HA-tagged subunits of the v-ATPase. However, it might be good to provide more details in the text. It is hard to appreciate the relevance of the blot shown since MW are missing (Fig 3B and C). Whole lysate indicates a problem of expression of some HA-tagged v-ATPase subunits. How does this affect the conclusions made by the authors? Temporality of the interaction? Why the other subunits do not appear in the immuno precipitate?
- 3) The reason for using Bmx +/- mice instead of Bmx-/- mice is not clear? Is the KO lethal? Is there a reason why these heterozygote mice were not uses in addition of BMX-IN1 in key experiments of Mtb burden.
- 4) Fig 6a and 6b are not convincing, and lack proper quantification. Since Bmx +/- are available, why the authors did not use them to assess ATP6V1E1 phosphorylation in response to Mtb? Finally, does BMX deciciency alter v-ATPase assembly at the lysosomal membrane?
- 5) In Fig 7, proper controls on PLGA-aNP on lysosomal pH stability are missing. How PLGA-aNP can end up in the same phagolysosomes as Mtb?
- 6) Although fluorescence intensity is widely use for lyso-Tracker-mediated evaluation of lysosomal acidification, I found this microscopy-mediated approach a little bit weak for this type of analysis (how many cells were used to determine fluorescence activity?)

Minor points:

- 1) Rephrase sentences in line 261-262, Line 276-277, and Lines 142-145. These sentences are particularly confusing.
- 2) English writing could be improved for clarity.
- 3) Is Rv1184c constitutively secreted by mtb or is it secreted at the time of infection? Add the MW and proper quantifications of all the blots.
- 4) Methods are poorly described.

Version 1:

Reviewer comments:

Reviewer #1

(Remarks to the Author)

The authors have taken my suggestions into account, and the manuscript is now suitable for publication.

(Remarks on code availability)

The authors have taken my suggestions into account, and the manuscript is now suitable for publication.

Reviewer #2

(Remarks to the Author)

I appreciate the extensive revision carried out by the authors. The authors have satisfactorily addressed most of my original comments.

I have the following comments on the experiment using FITC-labelled H37Rv, which the authors should address:

- Using FITC-labelled or any other labelled H37Rv has limitations. The best way is to use a GFP or mCherry-expressing Mtb strain. The protocol for FITC labelling is missing.
- In a few cells, large clumps of Mtb are visible. This could be due to the shedding of FITC or leftover clumps during single-cell suspension preparation. The presence of large clumps usually interferes with the quantification. How did the authors address this while analysing the data?

- Secondly, whether only confirmed infected cells from every field were used for quantifying the lysotracker staining?
- Finally, was there any effort to calculate the co-localisation of H37Rv, the Chp2-deleted strain and the complemented strain with lysotracker and any other lysosomal marker? This is important to demonstrate that WT Mtb, even if it ends up in the lysosome, blocks lysosomal acidification through Chp2.

The CFU experiment in BMDM is very good, but the authors have only presented it in the response letter. I wonder why this has not been included in either the main figures or in the supplementary information.

Reviewer #3

(Remarks to the Author)

The authors have adequately responded to the critiques of the reviews and have performed additional studies to support their conclusions. The key findings are convincingly established.

I have no additional concerns.

Point to point response

Reviewers' comments:

Reviewer #1: (Remarks to the Author):

In this article, the authors explore how *Mycobacterium tuberculosis* (Mtb) manipulates macrophages to inhibit lysosomal acidification and promote its own replication. They show that Mtb secretes the effector Rv1184c, which targets the V-ATPase acidification machinery, specifically the V1 subunit. A wide range of techniques was used, and substantial experimental data were generated to support their hypothesis. The study is well-constructed and very interesting. Mechanisms by which Mtb blocks lysosomal acidification remain poorly understood, making this work valuable. Furthermore, the *in vivo* inhibition of BMX appears to be a promising host-directed therapy (HDT) strategy.

Response: We are grateful to you for your favorable comments on our manuscript.

Below are some suggestions to improve the quality of the manuscript:

In the first section of the results, the authors conclude that the use of bafilomycin to inhibit V-ATPase demonstrates that Rv1184c inhibits lysosomal acidification via this pathway. This conclusion seems a bit premature and could benefit from an additional experimental approach to confirm the hypothesis, even if later experiments do support it further.

Response: We sincerely thank you for raising this important point. We fully agree that lysosomal acidification is a highly regulated process involving multiple coordinated mechanisms. Among these, V-ATPase is essential for lysosomal acidification, as it is the sole transporter that pumps protons into the organelle. Given this, we first investigated the role of the V-ATPase in the mechanism of lysosomal acidification regulated by Rv1184c.

The use of Bafilomycin A1 (Baf-A1), a specific inhibitor of V-ATPase, to inhibit lysosomal acidification and perform functional rescue experiments is a well-established and widely adopted strategy to validate whether a molecular mechanism acts through V-ATPase-mediated acidification[1]. Our results demonstrate that treatment with Baf-A1 rescues the lysosomal acidification defect phenotype caused by the H37Rv Δ Rv1184c mutant strain (**Fig. 1g**, please see below). This supports the conclusion that Rv1184c inhibits lysosomal acidification specifically via the V-ATPase pathway.

Note: Chp2 (Rv1184c)

In addition to the inhibitor experiments, we have also provided genetic evidence and specified subunit involved. We generated *Atp6v1e1* knockout mice and evaluated lysosomal acidification in macrophages infected with either H37Rv or H37Rv Δ Rv1184c. The results clearly indicate that the regulatory effect of Rv1184c on lysosomal acidification is dependent on the E1 subunit of the V-ATPase (Fig. 3q, r, please see below). The detail description of this analysis was provided in the revised results section (Line 200-205).

Note: Chp2 (Rv1184c)

Line 129: Infection in macrophages was monitored for 24 hours. This time point may be too early—what happens at later time points, such as 3 or 4 days post-infection?

Response: Thank you for this valuable comment regarding the infection time course. We agree that assessing later time points is crucial to ensure the observed phenotype is sustained and not a transient effect. In response to your suggestion, we extended the infection time of the intracellular survival assays to 4 days. The results consistently demonstrated that the increased clearance phenotype (reduced CFU) compared to the H37Rv was maintained at 96 hours for H37Rv Δ Rv1184c (Fig. 2a, b), confirming that the inhibitory effect of Rv1184c defect on the intracellular survival of *Mtb* observed at 24 hours is robust and representative. The new data (please see below) from the extended time course have been replaced the original data and figure legend have been updated in the revised manuscript.

Note: Chp2 (Rv1184c)

Line 131: Please explain the role of the NH₄Cl control.

Response: Thank you for raising this point. We apologize for the lack of clarity in our manuscript.

The NH₄Cl control was included to chemically inhibit lysosomal acidification and thereby block lysosomal function. As a weak base, NH₄Cl diffuses into acidic compartments (mainly lysosomes) and raises the intravascular pH, which disrupts the activity of acid-dependent hydrolases and impairs the degradative capacity of the lysosome[2-4].

The purpose of including this control was to serve as a positive functional benchmark for a lysosomal impairment phenotype. Our hypothesis was that if Rv1184c promotes bacterial survival primarily by compromising lysosomal function, then in macrophages with intact lysosomal function (untreated with NH₄Cl), the Rv1184c mutant, which preserves lysosomal function, should be cleared more efficiently than the wild-type strain. Artificially impairing lysosomes with NH₄Cl should phenocopy the effect of the Rv1184c protein and rescue the survival defect of the mutant. In other words, in NH₄Cl-treated macrophages where lysosomes are inhibited, the difference in survival between the wild-type bacteria and the Rv1184c mutant should be abolished.

Furthermore, this initial pharmacological evidence with NH₄Cl laid the groundwork for our subsequent, more specific investigations using both pharmacological (Baf-A1) and genetic (*Atp6v1e1*^{+/-}) inhibition of the V-ATPase, which collectively and robustly demonstrate that Rv1184c functions through this pathway.

We have added an explanation of the NH₄Cl control's function in the result section of the revised manuscript (**Line 139-141**).

The results show increased acidification during infection with the Rv1184c-deficient mutant strain, and that this strain is attenuated in macrophages. Is it possible to provide imaging data showing co-localization between bacteria and LysoTracker under these conditions? For instance, is there increased co-localization of the Rv1184c deletion mutant with LysoTracker compared to wild-type Mtb?

Response: Thank you for this excellent suggestion. We agree that providing direct imaging evidence for the increased lysosomal acidification and colocalization would greatly strengthen our conclusions.

As you recommended, we performed confocal microscopy experiments on macrophages infected with either wild-type H37Rv or the Rv1184c deletion mutant strains. Both bacterial strains were pre-labeled with FITC prior to infection. At 4 hours post-infection, the macrophages were stained with LysoTracker-Red to visualize acidic compartments. Consistent with our LysoTracker staining intensity data, we observed a significant increase in the colocalization of the Rv1184c mutant with LysoTracker-positive compartments compared to the wild-type bacteria (**Supplementary Fig. 1c, d**).

We have incorporated these new findings (please see below) in the revised manuscript, which provide direct visual and quantitative evidence that the Rv1184c deletion mutant is targeted to acidic lysosomes more frequently than the wild-type strain, which strongly supports our model that the Rv1184c protein functions to inhibit acidification.

Note: Chp2 (Rv1184c)

Line 135: Could you comment on the reduced immune cell infiltration? Which immune cell types are affected? Could this be assessed by flow cytometry?

Response: We thank you for this question regarding the specific immune cell types affected.

The observed reduction in overall immune cell infiltration is a key in vivo consequence of the enhanced bacterial clearance mediated by the Rv1184c deletion mutant. By promoting lysosomal acidification and degradation within macrophages during the critical early stage of infection, the mutant strain is contained more effectively. This superior early control leads to a lower overall bacterial burden and a subsequently attenuated inflammatory response in the lungs, which manifests as reduced recruitment of immune cells.

To identify the specific cell populations affected, we performed multiplex immunohistochemistry (mIHC) on the same lung sections from **Fig. 2g**. The results (**Supplementary Fig. 2c, d**) quantitatively demonstrate a significant reduction in both innate and adaptive immune cells, including CD11B⁺ monocytes/macrophages, MPO⁺ neutrophils, CD3⁺ T cells and CD20⁺ B cells, consistent with the resolution of a more effectively controlled infection.

Thank you for this insightful suggestion regarding the use of flow cytometry to assess immune cell infiltration. We agree that flow cytometry is an excellent method for immune cell profiling. In our study, we ultimately chose multiplex immunohistochemistry instead of flow cytometry primarily because it allowed us to use the same formalin-fixed, paraffin-embedded lung tissue sections from the same batch of animal experiments as those analyzed in **Fig. 2g**. This approach ensured sample source consistency and strengthened the comparability of results across different parts of the study.

We have incorporated these new findings (please see below) and detail description

of this analysis in the revised result section (**Line 146-148**).

Note: Chp2 (Rv1184c)

Line 145: Again, I am not fully convinced by the bafilomycin experiment. Inhibiting V-ATPase will necessarily favor infection and may mask the effect of the mutation. Could you consider an alternative readout?

Response: We sincerely appreciate the your constructive feedback. Bafilomycin A1 (Baf-A1) is a V-ATPase inhibitor. As such, employing Baf-A1 to inhibit lysosomal acidification for functional rescue experiments represents the most common strategy for validating whether a molecule exerts its role by regulating V-ATPase-mediated lysosomal acidification[5]. Based on our hypothesis that Rv1184c promotes *Mtb* intracellular survival by suppressing lysosomal acidification, we reasoned that Baf-A1 treatment should rescue the survival defect of the H37RvΔRv1184c mutant, restoring its bacterial burden (CFU) to a level comparable to that of the wild-type H37Rv infection (**Fig. 2d**).

Note: Chp2 (Rv1184c)

However, we also recognize that Baf-A1 treatment causes a global V-ATPase inhibition, thereby potentially obscuring the specific role of Rv1184c. Therefore, after identifying ATP6V1E1 as the specific V-ATPase subunit targeted by Rv1184c, we employed a more targeted genetic approach to substantiate our findings. We generated *Atp6v1e1*^{+/-} mice and evaluated both lysosomal acidification and intracellular bacterial survival in macrophages from these mice. The results demonstrated that the regulatory effect of Rv1184c on lysosomal acidification and

intracellular bacterial clearance is specifically dependent on ATP6V1E1(**Fig. 3q-s**), thereby confirming that Rv1184c exerts its function through this subunit of the V-ATPase complex.

Additionally, we recognize that lysosomal function is closely linked to multiple cellular processes-such as lipid metabolism, energy balance, and apoptosis/necrosis-all of which can directly or indirectly influence *Mtb* survival within host cells. Although we don't explicitly investigate whether these pathways are differentially affected, we indeed cannot exclude the possibility that Rv1184c-mediated suppression of lysosomal acidification may indirectly modulate these processes, thereby contributing to enhanced bacterial survival in vivo.

The detail description of this analysis was provided in the revised result section (**Line 200-207**).

Note: Chp2 (Rv1184c)

Is it possible to visualize or measure lysosomal acidification in alveolar macrophages from infected mice?

Response: Thank you for this excellent and challenging question. We agree that directly assessing lysosomal acidification in alveolar macrophages from infected mice would provide invaluable in vivo validation of our model. Unfortunately, using LysoTracker or pHrodo to measure lysosomal acidification require live-cell imaging. Our Biosafety Level 3 (BSL-3) laboratory is unable to perform this test. We sincerely appreciate your understanding and have acknowledged this limitation in the discussion section of the manuscript.

Line 180: The in vivo results are interesting. Have you considered infecting *Atp6v1E1* knockout mice with the Rv1184c mutant strain? It would be interesting to discuss the expected results.

Response: Thank you for this excellent suggestion. We agree that using *Atp6v1e1*^{-/-} mice infected with the Rv1184c mutant strain would be a fascinating and direct experiment to further solidify our proposed mechanism.

As per your suggestion, we infected *Atp6v1e1*^{-/-} mice with H37Rv or H37RvΔRv1184c to further validate in vivo relevance of Rv1184c and ATP6V1E1. The results showed knockout of *Atp6v1e1* markedly increased bacterial burden in lung tissues of the *Mtb* H37Rv-infected mice, and abolished the increased bacterial burden by Rv1184c in lung tissues of the *Mtb*-infected mice (**Fig. 3u, Supplementary Fig. 3p**), suggesting Rv1184c promotes intracellular survival of

Mtb in vivo by targeting ATP6V1E1. The new data (please see below) and analysis (**Line210-214**) were included in the revised result section.

Note: Chp2 (Rv1184c)

BMX inhibition shows very promising data. Could this therapy be combined with standard antibiotic treatment to test for potential synergy?

Response: We thank you for this excellent suggestion regarding potential combination therapy of BMX inhibitor. To investigate this, we treated *Mtb*-infected macrophages with the BMX inhibitor alone or in combination with rifampicin, the most commonly used first-line anti-tuberculosis drug. The results demonstrated this combination failed to enhance bactericidal efficacy against intracellular *Mtb* (please see below), suggesting there is no synergistic interaction between the BMX inhibitor and rifampicin. While we acknowledge the limited scope of drugs tested in combination with the BMX inhibitor, this does not rule out potential synergy with other anti-tuberculosis drugs, which we will explore in future work.

What would be the effect of BMX inhibition in mice infected with the Rv1184c mutant strain?

Response: We thank you for this excellent suggestion. To address this, we infected mice with either H37Rv or H37RvΔRv1184c. After infection two weeks, we initiated BMX inhibitor treatment and assessed the outcomes two weeks later. The results showed that the BMX inhibitor significantly reduced the bacterial burden and pathological damage in mice infected with wild-type H37Rv. Crucially, this therapeutic effect was abolished in mice infected with the H37RvΔRv1184c mutant

(Fig. 7f-i).

This finding indicates that the efficacy of the BMX inhibitor is dependent on the presence of the bacterial effector Rv1184c. It supports a model wherein the inhibitor acts by targeting the host pathway that is subverted by Rv1184c, thus neutralizing the bacterium's virulence strategy. This suggests that host-directed therapy (HDT) with BMX inhibitors could be particularly effective against *Mtb* strains that rely on this specific mechanism.

We have added these new data (please see below) and the corresponding analysis (Lines 370-384) to the revised manuscript.

Note: Chp2 (Rv1184c)

Similarly, what would be the effect of PLGA-aNP treatment in *Atp6v1E1* knockout)

Response: Thank you for this insightful suggestion. Investigating the effect of PLGA-aNP treatment in an *Atp6v1e1* knockout model would indeed be a fascinating experiment to further dissect the mechanism of action, and we appreciate you raising this idea.

During the revision process, and in an effort to sharpen the focus of the manuscript on the novel Rv1184c-BMX-ATP6V1E1 signaling axis we have identified, we made the decision to remove the PLGA-aNP treatment data. While these data were interesting, we agreed that its mechanism was not directly upstream or downstream of the Rv1184c-BMX-ATP6V1E1 pathway that is the central focus of this study, and its inclusion diluted the main narrative, which is also pointed out by reviewer 2#.

Although this specific experiment is not included in the current manuscript, we have carefully noted your suggestion regarding the *Atp6v1e1* knockout. It represents an excellent direction for a future study to understand how modulating lysosomal pH intersects with specific host pathways for antimicrobial activity. We sincerely appreciate your guidance on this point and plan to pursue it in our subsequent work.

Reviewer #2: (Remarks to the Author):

This manuscript by Chen et al, titled “Mycobacterium tuberculosis manipulates phosphorylation of ATP6V1E1 to promote its intracellular survival”, shows an important phosphorylation-dependent regulation of lysosomal acidification during Mtb infection, with the potential to regulate Mtb survival inside the infected macrophages. The authors show that Mtb secreted protein Rv1184c facilitates interaction between tyrosine kinase BMX and the vATPase subunit ATP6v1e1, leading to increased phosphorylation of ATP6v1e1 and corresponding decline in the lysosomal acidification, resulting in less efficient bacterial killing. In view of that, inhibiting BMX kinase in the animal models shows a decrease in lung CFU burden, which is dependent on the protein ATP6v1e1.

The regulation of lysosomal activity by Mtb during infection has been under immense focus for investigation due to its direct role in killing intracellular bacterial infections. The current manuscript provides interesting insights into this key battle between the host and the pathogen. The authors generate useful models to address the specific questions asked in this study.

Response: We appreciate your supportive assessment of our work.

However, the study suffers from some serious limitations. There are some general observations and specific concerns pointed out below.

General observations: There is a lack of details on the experiments and results. For studies on Mtb infection, cell/cell-lines used, the time points post-infection, duration of treatment, etc., are vital details. These details are missing even in the methods section. Some of the results reflect a less robust experimental strategy, leading to deviations from the expected line.

Different experiments are done on different cell lines/types. The rationale for those is not obvious. Cells used for the experiments are not explicitly mentioned.

Response: We thank you for this critical assessment and the opportunity to strengthen our manuscript. We sincerely apologize for the lack of essential details and the perceived lack of robustness in our experimental strategy.

We have thoroughly revised the manuscript to address these general concerns comprehensively. The changes are detailed below.

We have now extensively expanded the Methods section to include all missing critical details, including the complete cell line information, the precise time points, the treatment durations, and the step-by-step protocols. Furthermore, we have revised that all figure legends now explicitly state the specific cell type and key experimental conditions (e.g., MOI, time points).

We have revised the main text to clearly articulate the scientific rationale behind each cell model selection, demonstrating how their complementary use provides a more robust and physiologically relevant validation of our findings.

All the modifications mentioned above have been incorporated into the manuscript and are highlighted in red for easy identification. We believe they significantly

improve the manuscript's rigor and clarity.

Major limitations:

The ability of *Mtb* to block phagosome maturation and lysosomal acidification is well-documented. However, throughout this study, a significantly higher number of bacilli is seen in acidified lysosomes, which, of course, increases upon loss of Rv1184c. The observed phenotype is difficult to interpret due to the absence of any staining for the *Mtb*, which would allow assessing bacterial presence in acidified lysosomes.

Response: We thank you for this critical insight. We agree that bacterial staining is essential to definitively link the observed increase in lysosomal acidification to the location of the bacteria themselves.

As per your suggestion, we performed confocal microscopy with FITC-labeled bacteria and LysoTracker-Red in macrophages. We observed a significantly higher degree of colocalization between the FITC-labeled H37Rv Δ Rv1184 and the LysoTracker-positive compartments compared to the wild-type H37Rv (Supplementary Fig. 1c). In addition, we have re-examined the *Mtb* staining for the other lysosomal acidification (Supplementary Fig. 1d, Supplementary Fig. 3j, o and Supplementary Fig. 4b)

These new data directly confirm that the increased lysosomal acidification we reported previously is indeed associated with the bacilli themselves, and that loss of Rv1184c leads to a significantly higher proportion of bacteria being trapped in acidified lysosomes. This provides strong visual and quantitative support for our model that the Rv1184c protein is essential for *Mtb* to inhibit phagolysosomal maturation.

Note: Chp2 (Rv1184c)

Conclusions for a few results and interpretations made thereof are not aligned with the experimental observations.

Response: We thank you for this general comment regarding the alignment of our conclusions with the experimental data. We agree that maintaining this precision is critical for the scientific rigor of the manuscript.

We therefore have conducted a thorough, line-by-line review of the entire text to ensure all interpretations are strictly constrained by the data presented. This involved refining quantitative analyses, ensuring all controls are appropriately referenced, and calibrating the language in the text to ensure all interpretations are fully and clearly supported by the observations. We believe these revisions have significantly improved the accuracy and rigor of the manuscript. We are happy to provide a point-by-point response if you could kindly specify any particular instances where they perceived a misalignment, so that we may address them with utmost precision.

The phenotype of ATP6v1e1 knockout mice is not well explained. With a loss of lysosomal function, in these animals, even without infection, they should show lysosomal storage disorder phenotype.

Response: We sincerely thank you for emphasizing this critical point, which was raised both as a major concern in this section and again in the specific comments.

As rightly highlighted, the homozygous deletion of *Atp6v1e1* results in embryonic lethality, a finding consistent with the MGI database. Cyagen, which is a global provider of genetically modified rodent models, also reported *Atp6v1e1* complete knockout mice is embryonic lethality (official link: <https://www.cyagen.com/mouseatlas/S-KO-01171>). This phenotype underscoring the gene's essential role in development and lysosomal function, much like what has been observed with other core lysosomal genes such as *LAMP2* and *CSTD*.

Although complete knockout of *Atp6v1e1* results in embryonic lethality, *Atp6v1e1* heterozygous mice appear normal and do not exhibit lysosomal storage disorder phenotypes (**Supplementary Fig. 3g**). Detection of V1E1 protein expression in peritoneal macrophages of uninfected mice showed that the protein level in

heterozygous mice was significantly reduced, but approximately 50% of the wild-type level remained (**Supplementary Fig. 3e, f**). We propose that the reduction in protein dosage from a single allele is sufficiently compensated under basal conditions, likely through adaptive mechanisms that maintain minimal functional V-ATPase activity.

We are grateful for the opportunity to clarify this important matter and have revised the manuscript to include the new data (please see below) and a detailed discussion of these points (**Line 179-182**).

Specific points:

Question 1: How ATP6v1e1 overexpression alone is able to cause significant lysosomal acidification in 293T cells? The vacuolar ATPase is a multi-subunit protein, and the catalytic site, formed by the V1 domain, consists of all subunits. This needs more clarity.

Response: We sincerely thank you for this insightful question. You rightly points out V-ATPase is indeed a multi-subunit complex. Our data showed that among the individual V1 subunits overexpressed, only E1 subunit significantly enhanced lysosomal acidification. We propose the explanations for this specific effect of V1E1 as below:

The functional V-ATPase complex requires the assembly of subunits in a precise stoichiometry. Research by Mark A. Compton *et al.* using a yeast model demonstrated that Vma9p (subunit e) acts as a rate-limiting factor in V-ATPase assembly. Disruption of the yeast VMA9 gene prevents V₁ and V₀ subunits from assembling at the vacuolar membrane, while its overexpression enhances complex formation and boosts proton pump activity [6]. Vma9p is a small hydrophobic protein that is conserved from fungi to animals. Therefore, overexpression of the E1 subunit may indirectly enhance lysosomal acidification capacity by promoting the efficiency of V-ATPase assembly. Our findings indicate that phosphorylation of V1E1 can significantly influence the efficiency of V-ATPase assembly. We hypothesize that V1E1 overexpression, unlike other subunits, may bypass the normal stoichiometric constraints by acting as a rate-limiting assembly factor or a nucleating agent, thereby promoting the assembly of more functional complexes and enhancing proton pumping. We have added the analysis in the Discussion section of the revised manuscript (**Line 404-409**).

Question 2: What happens to ATP6ve1 expression upon Mtb infection, and whether that differs from the expression of other subunits?

Response: Thank you for raising this important question. As per your suggestion, we infected peritoneal macrophage with *Mtb* and detect the subunits expression. We found that the mRNA expression level of the ATP6V1E1 subunit did not change significantly at the time points post-*Mtb* infection that we examined. More importantly, the mRNA expression of other V-ATPase subunits (including other V1 domain subunits such as ATP6V1A, ATP6V1B2, ATP6V1C1, ATP6V1D, etc.) also showed no statistically significant differences before and after infection (please see below).

Together, these data indicate that, at the mRNA level and under our experimental conditions, *Mtb* infection does not significantly regulate the expression of ATP6V1E1 or other V-ATPase subunits. *Mtb* is unlikely to affect V-ATPase function via transcriptional modulation of these subunits.

Question 3: For lysosomal acidification during *Mtb* infection, there is no staining for *Mtb* (Fig. 1g, 3q, fig. S3g, etc.). The virulent *Mtb* is known to block lysosomal acidification through PMA. Since MOI used is 5, it is important to show that the cells analysed/being shown are also *Mtb*-infected.

Response: Thank you for this critical insight, which is also concerned by reviewer 1#. We agree that *Mtb* staining is important to confirm the analyzed cells are infected by *Mtb* and the increased lysosomal acidification is indeed associated with the bacilli themselves. As per your suggestion, we performed additional experiments to assess the co-localization of *Mtb* with lysosomes. This was achieved by pre-labeling *Mtb* with FITC and staining the cells with LysoTracker Red. *Mtb* staining for the other lysosomal acidification data was also re-examine. The new data demonstrate the presence of *Mtb* within the analyzed cell population (Supplementary Fig. 1c, d, Supplementary Fig. 3 j, o, Supplementary Fig. 4b), thus confirming that the measured lysosomal acidification occurs in infected cells.

This supplemental data strengthens our interpretation by directly demonstrating the co-localization of *Mtb* with acidified compartments in the analyzed cell population. We apologize for this omission in the original submission and are grateful for the opportunity to clarify this crucial point.

Note: Chp2 (Rv1184c)

Question 4: Why should H37Rv CFU decline in WT cells from 0 hour to 24 hours post-infection? This is a critical observation, *Mtb* CFU is generally stable or grows during 24 hours, however a decline in CFU indicates loss of virulence. Have the authors checked PDIM levels of the strains used? The observed dip in *Mtb* CFU in the WT cells needs to be analysed more robustly.

Response: Thank you for raising this important point. We appreciate the opportunity to clarify this question. Indeed, Jennifer A. Philips et al [7] and Dai Y et al [8] reported an increase in *Mtb* CFU in H37Rv-infected bone marrow derived macrophages(BMDMs) from WT mice for 48h or more. Ge P et al.[9] found *Mtb* CFU in H37Rv-infected U937 cells had increased within 0 to 24 hours. However, our data exhibited a decrease in *Mtb* CFU in H37Rv-infected primary peritoneal macrophages from mice at 24 hours. Consistent with our findings, several other

research using peritoneal macrophages have also reported a declining trend in H37Rv CFU post-infection[10] [11]. Thus, the difference of our data with some published literature could be due to infection of different type of macrophages. Furthermore, studies by Mosser, D. M., & Edwards, J. P.[12] have also confirmed that the bactericidal capacity of macrophages is associated with their specific type. This early CFU drop in peritoneal macrophages reflects the host's innate immune response, including phagolysosomal maturation and acidification, which successfully clears a portion of the intracellular *Mtb*. Subsequent bacterial replication, often observed after 24-48 hours, signifies the ability of surviving bacilli to overcome these host defenses. Therefore, the initial CFU decrease we observed is not indicative of a loss of virulence in our H37Rv strain but rather confirms the expected and validated response of competent macrophages to *Mtb* challenge.

Furthermore, we repeated the intracellular survival assay in BMDMs. The results showed an increasing trend in H37Rv CFU between 24 and 48 hours post-infection, while the CFU of H37Rv Δ Rv1184c remained relatively stable. Deletion of Rv1184c dramatically reduced the intracellular bacterial load compared to the H37Rv control (please see below), suggesting that Rv1184c promotes the intracellular survival of *Mtb* in BMDMs, consistent with the observations in primary peritoneal macrophages.

Note: Chp2 (Rv1184c)

Question 5: Tissue sections from uninfected ATP6v1e1 should also be shown. With the loss of lysosomal function, severe pathology may develop due to lysosomal storage disorder like condition.

Response: We thank you for this important suggestion. We agree that assessing basal pathology is crucial.

As we mentioned in the previous response, the homozygous *Atp6v1e1* knockout is embryonically lethal (MGI: 6148313), a definitive severe phenotype. Although we did not further investigate the exact cause of death, we speculate that it may be attributed to severe impairment of lysosomal function caused by the absence of ATP6V1E1. In our study, we used heterozygous *Atp6v1e1* deletion mice. As per

your suggestion, we examined lung sections from uninfected *Atp6v1e1*^{+/-} mice and found no signs of spontaneous pathology or storage disease, indicating that the partial gene dosage reduction is functionally compensated under baseline conditions (**Supplementary Fig. 3g**).

The phenotypic consequence of this haploinsufficiency is, however, unmasked under stress. We propose that the heightened lysosomal demand during Mtb infection exceeds this compensatory capacity, leading to the specific defects in acidification and bacterial clearance that we report. This clarifies that the *Atp6v1e1*^{+/-} model reveals a context-dependent vulnerability, not a constitutive disorder. We have added this data and discussion to the manuscript (**Lines 182-186**).

Line 178: LysoTracker staining shown in WT H37Rv infection is too high. H37Rv blocks lysosomal acidification. Secondly, fig. 3q-r: the lysoTracker staining upon H37Rv infection visible in the images at 0 hours and 4 hours in the WT and ATP6v1e1^{-/-} cells (3q) is not reflected in the corresponding quantification plot (3r). This is very concerning.

Response: We thank you for these critical observations, which allow us to clarify key aspects of our experimental system and data presentation.

Regarding the level of LysoTracker signal in H37Rv-infected cells, you are correct that H37Rv employs strategies to inhibit lysosomal acidification. However, the final lysosomal pH represents a dynamic equilibrium between the host's efforts to acidify the compartment and the bacterium's efforts to neutralize it. In our experimental conditions, the equilibrium pH remains within the acidic range detectable by LysoTracker. This is consistent with findings from other groups. For instance, **Gaur RL,et.al** [13] and **Sachdeva K et al.**,[14] have also reported positive LysoTracker staining in H37Rv-infected cells with similar fluorescence intensity.

Our model posits that the wild-type strain, secreting Rv1184c and other effectors, shifts this balance toward a more neutral pH. In contrast, the Δ Rv1184c mutant is defective in this inhibition, resulting in a failure to counteract host acidification and thus a higher observed LysoTracker signal. The signal in WT infection is therefore not artifactually high but reflects the outcome of this active host-pathogen struggle. Regarding the apparent discrepancy between the representative images in **Fig. 3q** and the quantification in **Fig. 3r**, we sincerely apologize for the confusion caused by the suboptimal presentation of the data in the original **Fig. 3r**. To address this

issue, we have revised the presentation of this panel. The revised figure (please see below) clearly shows that in WT macrophages, H37Rv Δ 1184 infection significantly promotes lysosomal acidification compared with H37Rv; however, this enhancing effect was attenuated in *Atp6v1e1*^{+/-} macrophages. Although the conclusion in the revised Fig. 3r remains consistent with the previous version, the data presentation has been improved and is now clearer. We have incorporated the revised figure into the updated manuscript. We sincerely thank you for this valuable suggestion, which has significantly improved the clarity and quality of our data presentation.

Note: Chp2 (Rv1184c)

Fig 6a: Rv1184c deletion: the impact on ATP6v1e1 phosphorylation is not very distinct from the WT infection.

Response: Thank you for your careful observation, which is highly valuable. During our assessment of ATP6v1e1 phosphorylation levels, we also simultaneously measured the level of ATP6V1E1 protein, although those results were not shown. To address this concern directly and unequivocally, we have conducted exact densitometric analysis of the phospho-ATP6V1E1 signal with normalization to total ATP6V1E1.

The quantified data demonstrated a decrease in ATP6V1E1 phosphorylation in macrophages infected with the Δ Rv1184c mutant strain compared to those infected with H37Rv (**Fig. 6a**). These results indicate that Rv1184c protein is indeed required for the full induction of ATP6V1E1 phosphorylation during *Mtb* infection. The quantitative data now provides clear support for our conclusion.

To avoid the limitations of visual assessment in evaluating ATP6V1E1 phosphorylation, we have provided the total ATP6V1E1 protein levels and supplemented all of the densitometric quantification data for the phospho-ATP6V1E1.

We have updated the text in the Results section to reflect this quantification (please see below). We are grateful to you for prompting us to strengthen this critical piece

of evidence.

a

Note: Chp2 (Rv1184c)

The Rv1484c knockout and complementation data is clean. However, authors need to show that sufficient Rv1484c gets secreted during infection in macrophages to be able to interact with and alter ATP6v1e1 phosphorylation. So far all such evidences are shown using overexpression experiments but whether expression and secretion of native protein is sufficient to alter ATP6v1e1 phosphorylation and lysosome acidification?

Response: Thank you for this critical question regarding the secretion and functionality of natively expressed Rv1184c. We agree that demonstrating this under physiological infection conditions is essential.

In our original submission, we included a Co-IP experiment (**Fig. 6j**) from macrophages infected with wild-type *Mtb* H37Rv, which showed that endogenous ATP6V1E1 co-precipitates with Rv1184c. This provided initial evidence for a native interaction.

To directly address your point and further strengthen this conclusion, we have now performed additional immunofluorescence experiments. As shown in the new data, staining with a specific anti-Rv1184c antibody visually confirms that the protein is secreted into the host cytoplasm during infection and partially co-localizes with LysoTracker, providing spatial confirmation of its translocation (**Supplementary Fig. 5a**).

Together, the Co-IP (**Fig. 6j**) and new IF data (**Supplementary Fig. 5a**) form a complementary and robust body of evidence, demonstrating that natively expressed and secreted Rv1184c is present at functionally relevant levels to interact with its host target. We have added this new data and analysis to the revised manuscript (**Line 344-346**) and are grateful for your suggestion, which has

significantly strengthened this part of our study.

Note: Chp2 (Rv1184c)

siBMX treatment: How many hours after BMX knockdown infection was given? What does 0 hour mean here? What is the knockdown efficiency of BMX siRNA? It is strange to see a complete loss of ATP6v1e1 phosphorylation.

Response: We thank you for these important questions, which allow us to clarify the experimental details and provide additional validation data. In our study, cells were transfected with siRNA targeting *Bmx* (or non-targeting control siRNA) for 48 hours to achieve optimal gene silencing prior to *Mtb* infection. The time point in the immunoblot analysis refers to the timing of *Mtb* infection in cells transfected with *Bmx* siRNA for 48 hours. The "0 hour" time point refers to cells transfected with *Bmx* siRNA for 48 hours but not infected with *Mtb*.

We apologize for not including the *Bmx* siRNA knockdown efficiency data from the original submission. In our study, we used a commercially available *Bmx* siRNA (product code sc-38942, as noted in the Method section) from Santa Cruz Biotechnology, Inc. It is a pool of three target-specific 19-25 nt siRNAs designed for gene knockdown and is a well-established and widely used reagent (Thomas, **J.D., et al.**[15]; **August, A., et al.**[16]). Upon acquisition of the *Bmx* siRNA, we subsequently assessed its knockdown efficiency at the mRNA levels. The results demonstrated that the knockdown efficiency exceeds 50% (**supplementary Fig. 4a**) (please see below).

a

We thank you for this keen observation for p-ATP6V1E1 blotting. We agree that the original blot for p-ATP6V1E1 in the siRNA-*Bmx* group appeared unexpectedly weak, likely due to suboptimal lane ordering and exposure conditions. We have since repeated the experiment with careful adjustments. The new, clear data consistently show a significant decrease in p-ATP6V1E1 upon *Bmx* knockdown, robustly supporting our initial conclusion (**Fig. 6b**) (please see below).

The consistency of this result-across both pharmacological inhibition (BMX-IN-1) and genetic knockdown (siRNA)-provides compelling and complementary evidence that BMX is the essential kinase for Mtb-induced ATP6V1E1 phosphorylation.

Note: Chp2 (Rv1184c)

For all phosphorylation immunoblots, GAPDH is not a correct loading control. The control should be the ATP6v1e1 protein itself.

Response: We thank you for emphasizing the importance of showing total ATP6V1E1 levels. We fully agree that showing the ATP6V1E1 protein itself is crucial for confirming the specific changes in its phosphorylation. We would like to clarify that the ATP6V1E1 protein levels were in fact measured in parallel in all relevant experiments and served as an internal control to confirm that the alterations in phosphorylation were not attributable to variations in total protein abundance. In the initial manuscript, to highlight the main findings, we primarily presented the phospho- ATP6V1E1 results and don't show the total ATP6V1E1 protein data.

Following your suggestion, we have added total ATP6V1E1 and performed densitometric quantification of the phospho-ATP6V1E1 signal with normalization to total ATP6V1E1 for each sample lane. The results confirm that total protein levels remain constant across conditions, solidifying our conclusion that the observed band intensity differences are due to specific phosphorylation changes. This revision has fully addressed the point and strengthened our manuscript.

The PLGA-aNP experiment is not relevant to the current study unless it is shown in combination with the BMX inhibitor.

Response: We thank you for this feedback. We agree with your assessment regarding the weak relevance of the PLGA-aNP (acid-sensitive nanoparticle) data with the current manuscript, which is firmly focused on elucidating the role of the Rv1184c-BMX-ATP6V1E1 axis in modulating lysosomal acidification.

The PLGA-aNP experiment was initially included as a broader conceptual proof that enhancing lysosomal acidification could be a potential therapeutic strategy. However, during the revision process, we realized its mechanism is distinct from the specific Rv1184c-BMX-ATP6V1E1 pathway that is the focus of this work, and its inclusion diluted the main narrative. We decided to remove it to maintain focus and prevent distraction from the core narrative.

Investigating the effect of PLGA-aNP combined with the BMX inhibitor would indeed be a fascinating experiment to further dissect the mechanism of PLGA-aNP action, and we appreciate you raising this idea. While those combination experiments would be more appropriate as a separate and extensive study. It represents a promising strategy to explore the potential synergistic effects of combining multiple HDT drugs that have proximate targets. Although this specific experiment is not included in the current manuscript, we sincerely appreciate your guidance and plan to pursue it in our subsequent work.

Minor points:

Typo in line 233: “numerous pathological pathways”?

Response: We sincerely thank you for their meticulous reading and for identifying this typographical error. The word “nuerous ” has been corrected to “numerous” in the revised manuscript.

We appreciate your attention to detail, which has helped improve the accuracy of our manuscript.

Authors should use Grammarly or similar software for written English.

Response: We sincerely thank you for this helpful suggestion. We apologize for any language-related issues in the initial submission.

In response, we have thoroughly revised the entire manuscript with a focus on improving the clarity, flow, and precision of the English language. We have enlisted the help of several colleagues proficient in academic English and utilized professional editing tools to refine the text.

We believe these revisions have significantly enhanced the readability of the manuscript, and we appreciate your suggestion, which has helped us improve the overall quality of our work.

Reviewer #3 (Remarks to the Author):

This manuscript by Chen J et al. uncovers a mechanism by which Mycobacterium tuberculosis H37Rv (Mtb) manipulates phagolysosomal maturation to avoid killing by macrophages. The authors show that Mtb protein Rv1184 promotes BMX interaction with the v-H⁺-ATPase subunit ATP6V1E1 thereby enhancing phosphorylation of ATP6V1E1 at tyrosine residues. This inhibits the v-ATPase assembly at the membrane, dampening phagolysosomal acidification and allowing Mtb survival. This manuscript therefore highlights an additional mechanism by which Mtb manipulate phagosome maturation.

Response: We appreciate your pertinent comments.

In its large part, I found this MS very interesting and convincing. It provides many details in the mechanism described with very neat experiments. My enthusiasm was somehow dampened by some flows in the logic that brought the authors to realized some experiments. For example, what brought the authors to assess phosphorylation fo ATP6V1E1 in the first place?

Response: We are sincerely grateful to you for your overall positive assessment of our work and for their meticulous reading of the manuscript. We also deeply appreciate them pointing out the specific issues, which we have addressed below. The decision to assess the phosphorylation of ATP6V1E1 was driven by the central and well-established role of phosphorylation as a primary mechanism for the rapid and reversible regulation of protein function in virtually all cellular processes, including signal transduction. While the V-ATPase is essential for general acidification, emerging evidence suggests that specific subunits can be subject to post-translational modifications, allowing for nuanced regulation. As we mentioned in the discussion section of the text, it has been reported that the phosphorylation of V-ATPase subunits affects lysosomal acidification. For example, the A subunit of V-ATPase is phosphorylated by protein kinase A (PKA), AMP-activated protein kinase (AMPK) or Aurora kinase A (AURKA), while the C subunit of V-ATPase is phosphorylated by WNK (with no lysine (K)) protein kinases[17-19]. This led us to assess phosphorylation of ATP6V1E1 in the first place. We add the analysis in the revised result section (**Line 227-228**).

Furthermore, there is a duplication of images in Fig 7b and 7g: 1) Mock controls are the same images but it is not mentioned in the legend ; 2) BMX-IN-1 #3 is the same image as PLGA-aNP #2.

Response: Thank you for your meticulous review of our manuscript. We are deeply grateful for the time and effort you have dedicated to it.

We sincerely apologize for this serious oversight in our figure preparation. You are absolutely correct. The duplication of the images was a serious mistake that occurred during the assembly of the figure panels. This does not reflect a lack of independent biological replicates for each condition but was an error in image selection and labeling. To correct this, we have thoroughly re-examined all our

original raw image data and identified the correct, independent images for each specific experimental condition (Mock, BMX-IN-1, and PLGA-aNP). We have generated a completely revised Figure 7 with the correct images in panels 7b and 7g. However, as one of the reviewers considered the PLGA-aNP results to be not closely related to the main narrative of the article, this section has been removed from the revised manuscript. To facilitate your review and verify the correctness of our revision, we provide the revised image data in the response letter (please see below).

We sincerely apologize for this oversight, which was unintentional. The corrected figure is now included in the revised manuscript (**Fig.7b**). We are grateful that this was caught during review, allowing us to correct the record and present an accurate representation of our data.

b

Thank you again for your time and for providing feedback that has significantly improved the clarity and accuracy of our manuscript.

I must say that my expertise in histological HandE staining in the context of Mtb infection is limited. Thus I could not really evaluate this part of the work.

Response: We thank you for this comment and sincerely apologize for the lack of systematic interpretation of the H&E staining results in our original manuscript, which limited the clarity of our pathological analysis.

To ensure our pathological analysis is rigorous and clear, we have now provided a detailed description of our scoring method in the Methods section (**Line742-745**). Briefly, pathology scores (0-10) were based on the ratio of inflammatory cell infiltration area to the total lung section area on the maximal cross-section of a lung lobe, with heavily stained areas indicating infiltration and damage, and lighter areas representing normal alveolar structure. Furthermore, to quantitatively characterize

the inflammatory infiltration, we performed multiplex immunohistochemistry for key immune cell markers: CD3 (T cells), CD20 (B cells), MPO (neutrophils), and CD11b (monocytes/macrophages) (**supplementary Fig. 2c, d**). The consolidated data robustly demonstrate that the deletion of Rv1184c limits the overall inflammatory infiltration in the host lungs.

Note: Chp2 (Rv1184c)

Some suggestions/comments that may improve the study:

Fig 1b, do the authors found CISH and PtpA in their screen?

Response: We thank you for raising the point about CISH and PtpA, two known inhibitors of phagosome acidification. Our initial screening plasmid library, while large-scale and validated in previous studies[10, 11], did not have 100% coverage of the *Mtb* secretome and unfortunately did not include CISH and PtpA. Importantly, our screening system was robust enough to identify other well-established regulators of lysosomal acidification, such as LpqH (Rv3763). This confirms the effectiveness of our approach, which successfully led us to identify Rv1184c as a novel player in this process. We have noted the library's coverage in the revised manuscript.

What is the effect of Rv1184c by itself upon injection in mice?

Response: We thank you for this question. To test the systemic effect of Rv1184c alone, we intravenously or intranasally injected purified protein into naive mice. Histopathological analysis revealed no significant lung damage compared to controls (**Supplementary Fig. 3q, r**) (please see below). This indicates that Rv1184c is not a general toxin but a precise effector whose major function, as our study shows, depends on its specific interaction with host ATP6V1E1 during infection. Again, we thank you for suggesting this experiment, which has provided valuable insight into the specific nature of this host-pathogen interaction.

q**r**
In Fig 3B, I understand that the authors over-expressed FLAG-Rv1184c together with HA-tagged subunits of the v-ATPase. However, it might be good to provide more details in the text. It is hard to appreciate the relevance of the blot shown since MW are missing (Fig 3B and C). Whole lysate indicates a problem of expression of some HA-tagged v-ATPase subunits. How does this affect the conclusions made by the authors? Temporality of the interaction? Why the other subunits do not appear in the immuno precipitate?

Response: We thank you for these insightful comments, which have allowed us to significantly strengthen this part of our study.

For the molecular weights and specificity, we apologize for the initial omission of molecular weight markers. In the revised manuscript, molecular weight markers have been added to all western blot results. Regarding the variable expression of the HA-tagged subunits of V1 domain, we have repeated the co-IP experiments with an expanded panel including all of the V1 subunits. The new data confirm that Rv1184c specifically co-immunoprecipitates with ATP6V1E1, but not with other V-ATPase subunits-including those that were robustly expressed. This underscores the specificity of the Rv1184c-ATP6V1E1 interaction, rather than a general association with the V-ATPase complex (**Fig. 3b**) (please see below).

b
For the temporality of the interaction, you raises a valuable point about the

dynamics of this interaction. We are pleased to clarify that this was already addressed in our original submission. The direct protein-protein interaction assay (**Fig. 3e**) and the endogenous co-IP during infection (**Fig. 6k**) provide compelling evidence that the Rv1184c-ATP6V1E1 interaction is specific and occurs in a physiological context, rather than being a transient or overexpression-induced artifact. We have incorporated these clarifications and new data into the revised manuscript.

Note: Chp2 (Rv1184c)

The reason for using *Bmx* +/- mice instead of *Bmx*-/- mice is not clear? Is the KO lethal? Is there a reason why these heterozygote mice were not used in addition of BMX-IN1 in key experiments of *Mtb* burden.

Response: We appreciate your comments. The generation of *Bmx* homozygous knockout (*Bmx*^{-/-}) mice for our experiments was not feasible due to challenges in obtaining a sufficient number of age-matched knockout and wild-type littermate controls from the breeding colony, largely attributable to low fertility. To enable a robust and practical investigation of *Bmx* function, we therefore utilized *Bmx*^{-/-} mice. Importantly, our results demonstrate that heterozygous loss of *Bmx* in macrophages is sufficient to produce a significant phenotype in enhanced lysosome acidification during H37Rv infection. This phenotype not only confirms the functional impact of reduced *Bmx* dosage but also aligns with and strengthens our conclusions from pharmacological or siRNA inhibition.

As per your suggestion, we further assessed the *Mtb* intracellular survival using primary peritoneal macrophages derived from *Bmx*^{+/-} mice. We found that heterozygous loss of BMX significantly enhanced the macrophages' capacity to clear *Mtb*. This result provides further evidence that BMX acts as a negative regulator of V-ATPase-mediated lysosomal acidification (**Fig. 5k, I**). We have

included this new data (please see below) in the revised manuscript.

k

l

Note: Chp2 (Rv1184c)

Fig 6a and 6b are not convincing, and lack proper quantification. Since *Bmx*^{+/-} are available, why the authors did not use them to assess ATP6V1E1 phosphorylation in response to *Mtb*?

Response: We thank you for raising the need for proper quantification. We would like to clarify that during our assessment of ATP6V1E1 phosphorylation levels, we also simultaneously measured the level of ATP6V1E1 protein, although those results were not shown. As per your suggestion, we have performed densitometric quantification of the phospho-ATP6V1E1 signal with normalization to total ATP6V1E1 for Fig 6a and 6b. This new quantitative analysis demonstrates that Rv1184c promotes the phosphorylation of ATP6V1E1 via BMX, which is consistent with our previous conclusions. We have updated the figures (**Fig. 6a, b**) (please see below).

a

b

Note: Chp2 (Rv1184c)

As per your suggestion, we assessed ATP6V1E1 phosphorylation in peritoneal macrophages from WT and *Bmx*^{+/-} mice. The new data (**Fig. 6c**) confirmed that the impaired ATP6V1E1 phosphorylation caused by Rv1184c deletion is abolished in *Bmx*^{+/-} macrophages. Critically, this genetic phenotype replicates the effect we observed with *Bmx* siRNA knockdown, powerfully confirming through an independent method that BMX is the primary kinase responsible for phosphorylating ATP6V1E1 in response to *Mtb* infection. This new data (please see below) was included in the revised manuscript.

Note: Chp2 (Rv1184c)

Finally, does BMX deficiency alter v-ATPase assembly at the lysosomal membrane?

Response: To determine if BMX-mediated phosphorylation of ATP6V1E1 impacts the assembly of the V-ATPase complex at the lysosomal membrane, we performed additional experiments as suggested by you. We analyzed lysosomal fractions from both BMX-IN-1 treated macrophages and *Bmx*^{+/-} macrophages, comparing them to their respective controls. Immunoblotting for key V1 (represented by ATP6V1E1) and V0 (represented by ATP6V0d1) subunits revealed that *Bmx* deficiency, whether achieved pharmacologically or genetically, reduces the abundance of these subunits at the lysosomal membrane (**Fig. 5f**). This result indicates that BMX is required for efficient V-ATPase assembly or stability at the lysosome, likely through its role in phosphorylating the ATP6V1E1 subunit. The new data (please see below) and the detail description (**Line286-293**) were included in the revised manuscript.

In Fig 7, proper controls on PLGA-aNP on lysosomal pH stability are missing. How PLGA-aNP can end up in the same phagolysosomes as Mtb?

Response: We thank you for this insightful comment and we agree that these are critical points to validate our drug delivery strategy. In our original results on PLGA-aNP, the lack of proper controls indeed prevents us from demonstrating their stability in response to lysosomal pH. To directly address this question, we

should employ bafilomycin A1 to inhibit lysosomal acidification and confirm the pH-dependency of PLGA-aNP drug release and efficacy. Additionally, we should conduct confocal microscopy studies using green fluorescently labeled PLGA-aNP and using LysoTracker Red to stain acidic compartments to detect co-localization between PLGA-aNP and acidic lysosomes, verifying their delivery to the intended compartment. These experiments will be a central part of our follow-up study.

Regarding your question on how PLGA-aNP can co-localize with *Mtb*, our understanding is as follows. The fate of most internalized particles, including our PLGA-aNP, is to undergo progressive maturation from early endosomes to late endosomes and finally to acidic phagolysosomes. This is a default and dominant pathway for phagocytosed material and is also a cell's innate clearance machinery. Even if a single *Mtb* bacterium successfully arrests its phagosome, the nanoparticles, being trafficked en masse via the cell's innate clearance machinery, have a high probability of being delivered to the same compartment. Of course, to directly address this point, we should perform additional experiment to provide direct visual evidence of their co-localization using labelled *Mtb*, PLGA-aNP and lysosomal markers.

The results of the PLGA-aNP were also concerned by the other two reviewers. One of the reviewers considered the weak relevance of the PLGA-aNP data with the current manuscript, which is firmly focused on elucidating the role of the Rv1184c-BMX-ATP6V1E1 axis in modulating lysosomal acidification. We decided to remove this section from the revised manuscript to maintain focus and prevent distraction from the core narrative.

Investigating the lysosomal pH stability of our PLGA-aNP and their co-localization with *Mtb* would indeed be a critical experiment to further validate our drug delivery strategy. Although this specific experiment is not included in the current manuscript, we sincerely appreciate your guidance and plan to pursue it in our subsequent work.

Although fluorescence intensity is widely use for lyso-Tracker-mediated evaluation of lysosomal acidification, I found this microscopy-mediated approach a little bit weak for this type of analysis (how many cells were used to determine fluorescence activity?)

Response: We thank you for raising this important point regarding the potential limitations of using fluorescence intensity-based microscopy for evaluating lysosomal acidification. We fully agree your opinion. In our original manuscript, to robustly corroborate our findings obtained with LysoTracker, we performed an additional, independent assay using pHrodo™-dextran. pHrodo™-dextran is a pH-sensitive probe that exhibits a strong increase in fluorescence specifically upon acidification within the lysosome, and its signal is less dependent on lysosomal loading compared to LysoTracker. This method is widely regarded as a more direct and quantitative measure of phagolysosomal acidification. Importantly, the results from the pHrodo™-dextran assay were entirely consistent with the trends observed in our LysoTracker experiments (**Supplementary Fig. 3k-l**) (please see blew). In

the original manuscript, we only briefly mentioned these confirmatory pHrodoTM-dextran results. In response to your insightful comment, we have now substantially

expanded the description of the pHrodoTM-dextran methodology in the results section (**Line189-192**) of the revised manuscript.

For all microscopy-based LysoTracker experiments, fluorescence intensity was quantified in a minimum of 40 individual cells from multiple randomly selected fields of view. This analysis was repeated across three independent biological replicates. The sample size (n) stated in the figure legends refers to the number of biological replicates (each containing data from >40 cells). We added the description in the revised method section (**Line629-633**).

Minor points:

Rephrase sentences in line 261-262, Line 276-277, and Lines 142-145. These sentences are particularly confusing.

Response: Thank you for this helpful suggestion. We agree that the indicated sentences were confusing and have revised them for clarity and precision. The changes are detailed below.

Revision for Lines 261-262:

Original: "Knockdown of *Bmx* abolished Rv1184c deletion induced inhibition of ATP6V1E1 phosphorylation."

Revised: " the inhibition of *Bmx* by siRNA reduced the difference in ATP6V1E1 phosphorylation between H37Rv and H37RvΔRv1184c-infected macrophages."

Revision for Lines 276-277:

Original: "By SPR assay, we found BMX has a strong affinity for ATP6V1E1 with a strong affinity (Kd=190 nM)."

Revised: "Surface plasmon resonance (SPR) measurements demonstrated that BMX binds ATP6V1E1 with a dissociation constant (Kd) of 190 nM, indicating a high-affinity interaction."

Revision for Lines 142-145:

Original: "Compared to the untreated group, Baf-A1 administration significantly attenuated the increase in Mtb survival in the lung induced by Rv1184c (Fig. 2k)."

Revised: " In H37Rv-infected mice, Baf-A1 treatment significantly increased the

bacterial load in the lungs compared to the untreated group, demonstrating that the clearance of *Mtb* in vivo is dependent on V-ATPase. Furthermore, V-ATPase inhibition by Baf-A1 abolished the difference in lung bacterial burden between H37Rv and H37Rv Δ Rv1184c infection (Fig.2k), indicating that Rv1184c enhances bacterial survival in mice by targeting the V-ATPase.

Thank you again for your careful review, which has greatly improved the clarity of our manuscript in blue highlight.

English writing could be improved for clarity.

Response: We thank you for this important feedback. We sincerely apologize for any lack of clarity in our original submission.

In response, we have thoroughly revised the entire manuscript with a focus on improving the clarity, flow, and precision of the English language. We have enlisted the help of several colleagues proficient in academic English and utilized professional editing tools to refine the text.

We believe these revisions have significantly enhanced the readability of the manuscript, and we appreciate your suggestion, which has helped us improve the overall quality of our work.

Is Rv1184c constitutively secreted by *mtb* or is it secreted at the time of infection?

Response: We thank you for the comment. In our study, Rv1184c was detected in both lysates and supernatants from cultures of H37Rv (**Supplementary Fig. 1a**), conforming that Rv1184c constitutively secreted by *Mtb*. To directly demonstrate its secretion during the infectious process, we performed a co-immunoprecipitation assay using macrophages infected with wild-type *Mtb*. We successfully immunoprecipitated endogenous host ATP6V1E1 and detected co-precipitating Rv1184c (**Fig. 6j**). To further provide spatial confirmation of its translocation, we conducted new immunofluorescence experiments. The results clearly showed secreted Rv1184c in the host cytoplasm, with partial co-localization with LysoTracker (**Supplementary Fig. 5a**), suggesting Rv1184c could be secreted by *Mtb* into macrophages during infection. The new data have added to the revised manuscript.

a

Note: Chp2 (Rv1184c)

Add the MW and proper quantifications of all the blots.

Response: We thank you for highlighting the need for complete blot data. We have now revised all immunoblot figures to address these omissions. Specifically, molecular weight markers have been added to each blot, and we have included densitometric quantifications above the corresponding bands.

Methods are poorly described.

Response: We appreciate your insightful suggestion, which is also concerned by reviewer 2#. We have comprehensively expanded the methods section to ensure full reproducibility, now including complete cell line specifications, exact experimental timepoints, detailed treatment protocols, and step-by-step procedures. The revised sections have been highlighted in red.

Reference:

1. Wang S, Tsun ZY, Wolfson RL, Shen K, Wyant GA, Plovanich ME, Yuan ED, Jones TD, Chantranupong L, Comb W *et al*: **Metabolism. Lysosomal amino acid transporter SLC38A9 signals arginine sufficiency to mTORC1.** *Science* 2015, **347**(6218):188-194.
2. Hamano T, Enomoto S, Shirafuji N, Ikawa M, Yamamura O, Yen SH, Nakamoto Y: **Autophagy and Tau Protein.** *Int J Mol Sci* 2021, **22**(14).
3. Jia L, Yu G, Zhang Y, Wang MM: **Lysosome-dependent degradation of Notch3.** *Int J Biochem Cell Biol* 2009, **41**(12):2594-2598.
4. Prabhakara C, Godbole R, Sil P, Jahnavi S, Gulzar SE, van Zanten TS, Sheth D, Subhash N, Chandra A, Shivaraj A *et al*: **Strategies to target SARS-CoV-2 entry and infection using dual mechanisms of inhibition by acidification inhibitors.** *PLoS Pathog* 2021, **17**(7):e1009706.
5. Yoshimori T, Yamamoto A, Moriyama Y, Futai M, Tashiro Y: **Bafilomycin A1, a specific inhibitor of vacuolar-type H(+)-ATPase, inhibits acidification and protein degradation in lysosomes of cultured cells.** *J Biol Chem* 1991, **266**(26):17707-17712.
6. Compton MA, Graham LA, Stevens TH: **Vma9p (subunit e) is an integral membrane V0**

- subunit of the yeast V-ATPase. *J Biol Chem* 2006, **281**(22):15312-15319.**
7. Mittal E, Prasad G, Upadhyay S, Sadadiwala J, Olive AJ, Yang G, Sasseti CM, Philips JA: **Mycobacterium tuberculosis virulence lipid PDIM inhibits autophagy in mice.** *Nat Microbiol* 2024, **9**(11):2970-2984.
 8. Dai Y, Zhu C, Xiao W, Huang K, Wang X, Shi C, Lin D, Zhang H, Liu X, Peng B *et al*: **Mycobacterium tuberculosis hijacks host TRIM21- and NCOA4-dependent ferritinophagy to enhance intracellular growth.** *J Clin Invest* 2023, **133**(8).
 9. Ge P, Lei Z, Yu Y, Lu Z, Qiang L, Chai Q, Zhang Y, Zhao D, Li B, Pang Y *et al*: **M. tuberculosis PknG manipulates host autophagy flux to promote pathogen intracellular survival.** *Autophagy* 2022, **18**(3):576-594.
 10. Liu S, Guan L, Peng C, Cheng Y, Cheng H, Wang F, Ma M, Zheng R, Ji Z, Cui P *et al*: **Mycobacterium tuberculosis suppresses host DNA repair to boost its intracellular survival.** *Cell Host Microbe* 2023, **31**(11):1820-1836.e1810.
 11. Peng C, Cheng Y, Ma M, Chen Q, Duan Y, Liu S, Cheng H, Yang H, Huang J, Bu W *et al*: **Mycobacterium tuberculosis suppresses host antimicrobial peptides by dehydrogenating L-alanine.** *Nat Commun* 2024, **15**(1):4216.
 12. Mosser DM, Edwards JP: **Exploring the full spectrum of macrophage activation.** *Nat Rev Immunol* 2008, **8**(12):958-969.
 13. Gaur RL, Ren K, Blumenthal A, Bhamidi S, González-Nilo FD, Jackson M, Zare RN, Ehrt S, Ernst JD, Banaei N: **LprG-mediated surface expression of lipoarabinomannan is essential for virulence of Mycobacterium tuberculosis.** *PLoS Pathog* 2014, **10**(9):e1004376.
 14. Sachdeva K, Sundaramurthy V: **The Interplay of Host Lysosomes and Intracellular Pathogens.** *Front Cell Infect Microbiol* 2020, **10**:595502.
 15. Thomas JD, Sideras P, Smith CI, Vorechovský I, Chapman V, Paul WE: **Colocalization of X-linked agammaglobulinemia and X-linked immunodeficiency genes.** *Science* 1993, **261**(5119):355-358.
 16. August A, Gibson S, Kawakami Y, Kawakami T, Mills GB, Dupont B: **CD28 is associated with and induces the immediate tyrosine phosphorylation and activation of the Tec family kinase ITK/EMT in the human Jurkat leukemic T-cell line.** *Proc Natl Acad Sci U S A* 1994, **91**(20):9347-9351.
 17. Al-bataineh MM, Gong F, Marciszyn AL, Myerburg MM, Pastor-Soler NM: **Regulation of proximal tubule vacuolar H(+)-ATPase by PKA and AMP-activated protein kinase.** *Am J Physiol Renal Physiol* 2014, **306**(9):F981-995.
 18. Al-Bataineh MM, Alzamora R, Ohmi K, Ho PY, Marciszyn AL, Gong F, Li H, Hallows KR, Pastor-Soler NM: **Aurora kinase A activates the vacuolar H+-ATPase (V-ATPase) in kidney carcinoma cells.** *Am J Physiol Renal Physiol* 2016, **310**(11):F1216-1228.
 19. Hong-Hermesdorf A, Brück A, Grüber A, Grüber G, Schumacher K: **A WNK kinase binds and phosphorylates V-ATPase subunit C.** *FEBS Lett* 2006, **580**(3):932-939.

Point to point response

Reviewers' comments:

Reviewer #1: (Remarks to the Author):

The authors have taken my suggestions into account, and the manuscript is now suitable for publication.

Reviewer #1 (Remarks on code availability):

The authors have taken my suggestions into account, and the manuscript is now suitable for publication.

Response: Thank you for the positive feedback and for your valuable suggestions throughout the review process.

Reviewer #2 (Remarks to the Author):

I appreciate the extensive revision carried out by the authors. The authors have satisfactorily addressed most of my original comments.

Response: Thank you for the positive feedback. We gratefully appreciate for your valuable suggestions.

I have the following comments on the experiment using FITC-labelled H37Rv, which the authors should address:

- **Using FITC-labelled or any other labelled H37Rv has limitations. The best way is to use a GFP or mCherry-expressing Mtb strain. The protocol for FITC labelling is missing.**

Response: Thank you very much for raising this insightful point. We completely agree with your perspective that using Mtb strains stably expressing GFP or mCherry is the "gold standard" for intracellular localization and long-term tracking studies, as it effectively avoids issues such as dye detachment or quenching.

In our study, we successfully constructed a GFP-expressing H37Rv strain. However, as the H37Rv Δ Chp2 strain was constructed by Shanghai Gene-optimal Science & Technology Co., we unfortunately did not obtain a GFP-expressing Chp2 knockout strain. To ensure consistency in the labeling method across all strains, we opted for FITC labeling. Fortunately, we have mastered this technique and have utilized it in previous studies [1]. In the staining phase, we strictly controlled the dye concentration and staining duration to ensure optimal labeling. In the washing phase, thorough PBS washes for four or five times were performed to maximally remove non-phagocytized bacteria and free dye attached to the cell surface. In the image analysis phase, any diffuse, non-particulate fluorescent background was excluded from quantitative analysis. Through rigorous quality control of the FITC labeling method, we ensured the reliability of our research conclusions.

We apologize for omitting the specific steps for FITC labeling in the initial manuscript. We have now supplemented the protocol for FITC labeling in the Methods section (**Line 549-560**) and as detailed below.

FITC labeling of Mtb

Mid-log phase Mtb cultures were harvested by centrifugation, washed twice, and resuspended in pre-chilled 0.1 M carbonate-bicarbonate buffer (pH 9.0). FITC isomer I was added to a final concentration of 0.1 mg/mL, and the suspension was incubated at 37°C for 1 hour in the dark with gentle agitation. After labeling, bacteria were washed thoroughly 4-5 times with pre-chilled phosphate-buffered saline (PBS, pH 7.4) until the supernatant was fluorescence-free to remove unbound dye. Finally, the labeled bacteria were resuspended in appropriate infection medium and quantified using McFarland standards coupled with colony-forming unit (CFU) plating. All labeling and infection steps were performed under light-protected conditions. In the image analysis phase, fields containing visible large clumps (>5 µm in diameter or containing clearly fused bacterial signals) were systematically excluded from quantification.

• In a few cells, large clumps of Mtb are visible. This could be due to the shedding of FITC or leftover clumps during single-cell suspension preparation. The presence of large clumps usually interferes with the quantification. How did the authors address this while analysing the data?

Response: We sincerely thank the reviewer for raising this important technical point regarding the presence of large Mtb clumps and their potential impact on quantification accuracy. We fully agree that large bacterial clumps can severely interfere with quantitative analysis, as they may lead to misinterpretation of fluorescence signals. In our study, fields containing visible large clumps (>5 µm in diameter or containing clearly fused bacterial signals) were systematically excluded from quantification. Only fields with well-dispersed, single-cell or small-cluster bacterial signals were analyzed. We have added this exclusion criterion to the Methods section of the revised manuscript (**Line 558-560**).

• Secondly, whether only confirmed infected cells from every field were used for quantifying the lysotracker staining?

Thank you for raising this important concern. We agree that, in the absence of fluorescently labeled bacteria, it is technically difficult to unambiguously identify every infected macrophage based solely on LysoTracker staining and bright-field microscopy, particularly at MOI of 5 where the infection rate cannot be precisely inferred at the single-cell level.

For this reason, in the revised version of the manuscript, we performed additional experiments using FITC-labeled Mtb to explicitly define infected cells and to quantify lysosomal acidification specifically within these cells. This approach allowed us to directly address the reviewer's concern and to rigorously validate our conclusions under conditions where infection status could be unambiguously determined.

Importantly, the infected-cell-restricted analysis yielded results fully consistent with our original observations, demonstrating that deletion of Chp2 enhances lysosomal acidification in infected macrophages. These data are now included in

Supplementary Fig1d (please see below) and form an integral part of the revised manuscript.

• Finally, was there any effort to calculate the co-localisation of H37Rv, the Chp2-deleted strain and the complemented strain with lysotracker and any other lysosomal marker? This is important to demonstrate that WT Mtb, even if it ends up in the lysosome, blocks lysosomal acidification through Chp2.

Response: Thank you for raising this insightful point. We fully agree that directly quantifying the co-localization of bacteria with lysosomes and with LysoTracker provides key evidence to support the hypothesis that Chp2 functions within lysosomes to block acidification.

Accordingly, we examined the co-localization of H37Rv, H37RvΔChp2, and the complemented strain with lysosomes using the canonical late-endosomal/lysosomal marker LAMP1, as well as their effects on lysosomal acidification using LysoTracker. The results showed that all strains showed lysosomal co-localization without significant differences (**Supplementary Fig. 1e, f**). In parallel, H37RvΔChp2 notably enhanced lysosomal acidification compared to H37Rv, whereas complementation restored the acidification blockade (**Supplementary Fig. 1g, h**), in line with our previous conclusion. Together, all three strains reach a LAMP1-positive compartment to a similar significant extent, but only the H37Rv and complemented strains, possessing functional Chp2, can subsequently prevent lysosomal acidification, demonstrating that Mtb inhibits lysosomal acidification via Chp2.

We believe this integrated analysis robustly demonstrates wild-type Mtb trafficks to a lysosomal compartment, and its Chp2 is specifically responsible for blocking the acidification of that compartment. We have incorporated these new findings (please see below) and detail description of this analysis in the revised result section (**Line 107-114**). We thank the reviewer for the perceptive suggestions, which have significantly enhanced the rigor and depth of our work.

The CFU experiment in BMDM is very good, but the authors have only presented it in the response letter. I wonder why this has not been included in either the main figures or in the supplementary information.

Response: Thank you for your positive feedback on the CFU data in BMDMs and for this excellent suggestion. We agree that these results are crucial for supporting our findings on bacterial survival. Accordingly, we have now included this dataset as a new panel in **Supplementary Fig.2b** of the revised manuscript (**Line135-136**). The corresponding figure legend and Methods (**Line 576-578, Line 692-694**) have been updated accordingly.

Reviewer #3 (Remarks to the Author):

The authors have adequately responded to the critiques of the reviews and have performed additional studies to support their conclusions. The key findings are convincingly established.

I have no additional concerns.

Response: We are grateful for your constructive feedback.

Reference

- 1 Zheng R. *et al.* A genome-wide association study identified PRKCB as a causal gene and therapeutic target for Mycobacterium avium complex disease. *Cell Rep Med.* **6**:101923 (2025).